## Registered report

psychology

statistical reasoning, power, online training, sample size neglect

**Author for correspondence:**
D. V. M. Bishop
e-mail: dorothy.bishop@psy.ox.ac.uk

# Can we shift belief in the 'Law of Small Numbers'?

D. V. M. Bishop[1], Jackie Thompson[1,2] and
Adam J. Parker[1,3]

[1]Department of Experimental Psychology, University of Oxford, Anna Watts Building, Woodstock Road, Oxford OX2 6GG, UK
[2]School of Psychological Science, University of Bristol, The Priory Road Complex, Priory Road, Clifton BS8 1TU, UK
[3]Department of Experimental Psychology, Division of Psychology and Language Sciences, University College London, London WC1H 0AP, UK

 DVMB, 0000-0002-2448-4033; JT, 0000-0003-2851-3636;
AJP, 0000-0002-1367-2282

'Sample size neglect' is a tendency to underestimate how the variability of mean estimates changes with sample size. We studied 100 participants, from science or social science backgrounds, to test whether a training task showing different-sized samples of data points (the 'beeswarm' task) can help overcome this bias. Ability to judge if two samples came from the same population improved with training, and 38% of participants reported that they had learned to wait for larger samples before making a response. Before and after training, participants completed a 12-item estimation quiz, including items testing sample size neglect (S-items). Bonus payments were given for correct responses. The quiz confirmed sample size neglect: 20% of participants scored zero on S-items, and only two participants achieved more than 4/6 items correct. Performance on the quiz did not improve after training, regardless of how much learning had occurred on the beeswarm task. Error patterns on the quiz were generally consistent with expectation, though there were some intriguing exceptions that could not readily be explained by sample size neglect. We suggest that training with simulated data might need to be accompanied by explicit instruction to be effective in countering sample size neglect more generally.

# 1. Introduction

Compared with laypeople, scientists receive extensive training to help them understand and appropriately address the uncertainty of evidence. Yet many scientists fall short in their understanding of statistical concepts. One cognitive bias demonstrated by Tversky & Kahneman [1] is the 'belief in the

**Figure 1.** Six independent samples of simulated male height, each with sample size of 10 (*a*) or 60 (*b*). Each point represents the height of one male, drawn from a population with mean 178 cm and s.d. 10 cm, and the red bar is the mean for that sample. The horizontal blue line represents the population mean.

law of small numbers'. This refers to the tendency to overestimate the stability of estimates that come from small samples—which, following Yoon *et al.* [2], we shall term 'sample size neglect'. For instance, as shown in figure 1, if repeatedly sampling 10 men from a population, the mean height of the sample will be far more variable than when repeatedly sampling 60 men. People understand that sample size does not affect the expected mean value, but they tend not to appreciate that it has a large effect on the standard error of the mean (i.e. variability of the red bars). This has implications for understanding of statistical power, i.e. the numerical relationship between sample size and ability to detect a true effect. Sample size neglect can help explain why so many studies in psychology, and indeed many other scientific disciplines, are underpowered.

More than half a century ago, Cohen [3] embarked on a project of improving psychologists' understanding of statistical power, providing tools to help people compute power and documenting the extent of underpowered studies in social psychology, with the aim of reducing waste in research efforts. He analysed 70 studies published in the *Journal of Abnormal and Social Psychology* and found that mean power to detect small effects was 0.18, to detect medium effects was 0.48 and to detect large effects was 0.83. Given that most effects in this field are small or medium, this indicated serious limitations of study design. However, 27 years later, Sedlmeier & Gigerenzer [4] reported that things had not changed at all. And in 2016, similar conclusions were drawn from a large review of studies in social and behavioural sciences up to 2011 [5]. In 2018, a review concluded that low power remains a major factor explaining poor replicability of highly cited studies in psychology [6]. A wide range of areas are affected, from neuroscience [7] to infancy research [8]. Why, despite years of attempts to improve the dismal record of low power in psychology, do researchers persist in performing underpowered studies? We suspect the explanation may go beyond lack of training and reflect the influence of sample size neglect, which leads us to have intuitions about sample size that are at odds with reality. Consider this example [1]:

*Suppose you have run an experiment on 20 participants and have obtained a significant result which confirms your theory (z = 2.23, p < 0.05, two-tailed). You now have cause to run an additional group of 10 participants. What do you think the probability is that the results will be significant, by a one-tailed test, separately for this group?*

Tversky and Kahneman reported that the majority of researchers who responded to this question were wrong in stating that the probability is somewhere around 0.85, while only 9 out of 84 researchers gave a more accurate answer (i.e. between 0.40 and 0.60). This is thought to reflect inaccurate beliefs in sampling that have unfortunate consequences in the course of scientific inquiry. Researchers view randomly drawn samples as highly representative of the population to a greater extent than theory predicts, at least for small samples.

If we are to tackle the problem of wasteful and misleading underpowered studies, we need to find ways to overcome sample size neglect. There is some evidence that debiasing training can help: Yoon *et al.* [2] showed improvements on a scale measuring this construct after either direct instruction, direct playing of a game designed to train awareness, or observing another person play the game. However, the scale items assessed rather broad aspects of generalizing from a few instances, e.g. respond on a 7-point scale: '*Micah's 10-year-old daughter Felicia scores two goals in her very first soccer*

*game. Based on this, Micah proudly predicts that Felicia will be the top scorer for her team for the year (25 games). How confident are you in Micah's prediction?'.* Our goal was to go beyond a general appreciation of the dangers of relying on small samples to help scientists obtain a more intuitive understanding of statistical power.

Informally, we have found that exposing students to simulations that allow them to visualize the variation between samples of different sizes can help counteract over-reliance on small samples to evaluate hypotheses. By generating datasets with known effect sizes and drawing random samples and subjecting them to statistical tests, students can learn to appreciate the ease with which we miss a true effect if the sample size is small. Simulations are a core aspect of a course outlined by Steel *et al.* [9] that trains statistical thinking in undergraduates, using simulations to help interpret patterns in data and to evaluate statistical power. Although their course appeared successful in training statistical thinking in the long term, they did not specifically evaluate the utility of simulations in counteracting sample size neglect. That is the goal of the current study.

In addition to training, simulations have been used to examine the decisions made by researchers. Morey & Hoekstra [10] used simulations to examine scientists' understanding of the logic of significance testing. In their task, scientists were asked to perform a series of experiments to judge which two groups of elves could make more toys based on a randomly assigned group difference between 0 and 1 standard deviations. Sample size and the test statistic were not shown to participants: instead they were shown displays that represented numerical information in terms of colour and location, with an opportunity also to view 'random shuffle' displays where they were told there was no true effect. Of the 136 participants for whom the null hypothesis was true, 86% correctly indicated no effect. When there was a true effect, correct decisions increased as a function of effect size. When the effect size was 0.3, accuracy was approximately 80%. For larger effect sizes, accuracy approached ceiling. When asked about the heuristics that they applied, 72% indicated using strong significance testing strategies. Thus, Morey and Hoekstra highlighted the potential utility of simulation-based training to answer questions about how researchers use information. However, their study only indirectly addressed sample size neglect, because participants were not told what the sample size was, and were encouraged to explore the sampling distribution for means when the effect size was zero, which made it relatively easy to see whether a given point fell within that distribution. Our focus is rather on the more lifelike situation when the participant knows the sample size and has to consider whether an observed distribution of scores is more likely to have come from a population with a true or null effect. As well as evaluating the efficacy of training on statistical judgements, we consider whether participants show evidence of adopting specific heuristics, such as waiting for a given sample size before making a judgement, or simply basing responses on observed effect size.

## 2. Online training task (beeswarm task)

To explore whether exposing people to simulated datasets can help develop a more intuitive sense of the relationship between variability of estimates and sample size, we created an online task that mimics the real-world process of gathering and interpreting data in the life and social sciences. In this task, participants (i.e. scientists) visually compare the distribution of simulated datasets and assess whether the samples come from a population where there is either a true effect (samples drawn from populations with differing means) or null effect (samples drawn from the same population). Potentially, the size of the effect in the population can be experimentally manipulated, but for the current study, we focus on cases where the true effect size is Cohen's $d = 0.3$ (equivalent to $r = 0.15$). This effect size was selected because it is, on the one hand, fairly typical of the kind of effect size obtained in many areas of psychology, biomedicine [11] and education [12], and on the other hand, there is a dramatic increase in the confidence with which results can be interpreted as favouring the true or null hypothesis as sample size increases (figure 2).

Participants were initially presented with underpowered samples: they could either respond immediately (true effect or no effect) or wait to see the sample size increase. Participants' subjective uncertainty was indexed by how long they wait to see more data. Ultimately, this paradigm not only measures perceptions of uncertainty; it is designed to help participants develop an intuitive sense of the uncertainty underlying even convincing-looking mean differences from small sample sizes.

We also examined the potential of this task to enhance participants' statistical reasoning by administering a pre- and post-training quiz, testing statistical reasoning using questions that assess sample size neglect. The quiz also included questions testing understanding of basic probability,

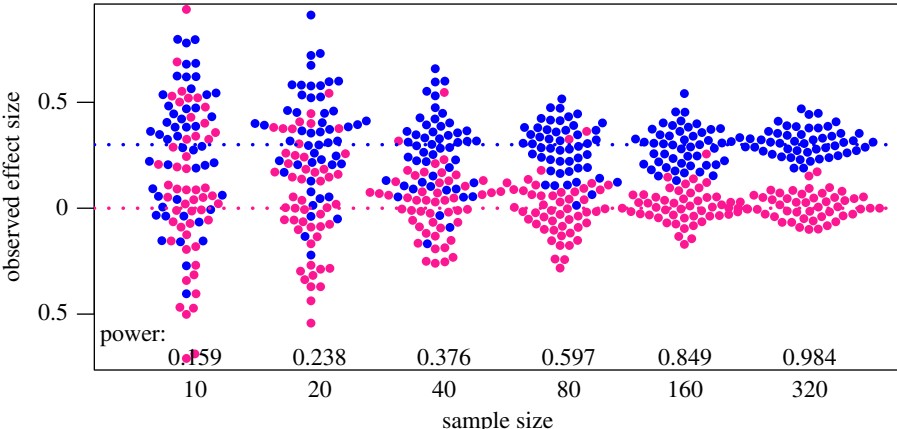

**Figure 2.** Simulated mean scores from samples of varying size, drawn from populations with either a null effect (pink) or a true effect size, Cohen's *d*, of 0.3 (blue). Power is the probability of obtaining $p < 0.05$ on a one-tailed *t*-test comparing group means for each sample size.

which act as positive controls: if participants score below 4/6 correct on the positive control questions, then this would indicate they either have little training in statistics, or are not taking the task seriously.

We pre-registered five predictions (see below) from a three-step hypothesis:

(a) Even people who have a reasonable grasp of probability theory suffer from sample size neglect.
(b) This neglect can be ameliorated by providing training that involves exposure to different sample sizes drawn from a population.
(c) Training will change understanding of how sample size affects accuracy of estimates, and this will be evident beyond the beeswarm task.

# 3. Method

## 3.1. Participants

Criteria for participants were (i) age of 18 years or over, (ii) have studied life or social sciences for at least one term at undergraduate (bachelor's degree) level. We specified that we would recruit up to 100 participants via the online research platform Prolific (www.prolific.co) and social media platforms. The maximum sample size was determined from a power calculation (see section on Simulated data for sample size determination, below) indicating that this is sufficient to detect the case when 33% or more of participants improve their performance on the beeswarm task.

## 3.2. Procedure

After providing informed consent, participants completed the estimation and judgement quiz online at a time and place of their choosing, followed (after training) by an alternative form of the post-training quiz. Participants were paid £7.50 for their time, but could earn a further bonus linked to the number of correct items on the quiz (3p per correctly answered item on each version) and their score on the training game to bring the payment up to a theoretical maximum of £12.

Training was divided into four blocks, and after each block, participants were told that they could take a break if they wished. The study was implemented using Gorilla [13], a cloud-based research tool for behavioural studies. The study took around 1 h to complete.

## 3.3. Design

Within-task learning was assessed in a within-subjects design with one factor with two levels (block of trials, comparing the first and last blocks, each with 20 trials). To avoid any confound between specific training items and sequence in training, a predetermined set of items was presented in new random order for each participant.

In addition, transfer of training was assessed using the estimation quiz, with parallel forms administered in a within-subject pre-/post-test design.

### 3.3.1. Judgement and reasoning quiz

The quiz (Appendix 1) was designed for this study and was presented to participants as a 'judgement and reasoning quiz' to minimize negative reactions from those who might feel they are poor at statistics. It has two parallel forms, each of which contains 12 multiple-choice items with four choices each. Six items test knowledge of general probability (P-items) and six test knowledge of how sample size determines accuracy (S-items). The parallel forms were counterbalanced, so half the participants received form 1 at pre-test and form 2 at post-test, and half received the opposite order. The main dependent variable is the number of items correct for S-items, but the pattern of error responses is also reported. A previous version of the quiz was piloted with 21 participants (Pilot 1), confirming that the difficulty level was appropriate, avoiding ceiling effects. On the basis of pilot testing and suggestions of reviewers, some items were reworded for clarity and to make S-items more relevant to the training.

### 3.3.2. Independent and dependent variables

On the beeswarm task, the independent variable is block: there are four blocks each of 20 items, but our focus was on comparing blocks 1 and 4. The dependent variables are (i) earnings in the game, which reflects both proportion correct for each block, and ability to select the optimal array index at which to respond (see below); (ii) mean array index per block, 1 to 6, corresponding to the array size (10, 20, 40, 80, 160 and 320 per group) at which the response is made. In pilot testing, we found that the earnings score is highly correlated with percentage correct ($r = 0.989$).

For analysis of self-reported response strategies, we use the comments in the free text-box asking about specific strategies adopted in the beeswarm task, which were coded by two independent raters as a binary variable: does or does not mention how accuracy is higher if they either wait for more information or focus on larger array sizes. This classification can be used as an independent variable to predict learning gain. On the quiz, the independent variable is session (pre-training or post-training) and the dependent variable is proportion correct for S-items and P-items.

### 3.3.3. Online training

**Instructions**: After the estimation quiz pre-test, participants were introduced to the game, as follows:
*Imagine you are a researcher running a large range of experiments. All of the experiments are comparing a control group with a group receiving an experimental treatment (intervention) of some sort. You are testing whether the experimental group has a higher score than the control group after the intervention/treatment.*

*For our purposes, it doesn't matter too much what the research topic is—the same principle applies to many different types of experiments. E.g. you could be examining:*

— *children's test scores after an educational intervention versus no intervention*
— *mouse weight after taking a drug versus a placebo*
— *changes in cholesterol levels in people following an experimental diet versus no diet*
— *effects of a chemical substrate on bacterial cell growth, compared to a control substrate (each data point measures growth of one cell culture)*

*You can choose which of these topics is most relevant to you and keep that example in mind.*

*Here we assume data in these trials to be normally distributed—i.e. the distribution of scores will fit a bell curve, with most data points close to the middle, and fewer data points at the extremes of very high or very low values.*

*The onion shapes to the right are both (sideways) normal distributions: the width of the shape represents the proportion of data points in this population with a given value on the y-axis. The boxplot in the centre of the plot shows information about the mean and variance of distributions for each condition.*

*The top image on the right compares two distributions separately; the bottom makes a side-by-side comparison. Feedback in this game will be shown like the bottom image.*

*If there is no effect of the experimental treatment, the overall distribution of the data will look like the figure on the bottom right, with the distribution of scores for the experimental group looking very similar to that of the control group.*

*In this game, you will see data from experimental trials. On 50% of trials, the data come from a population where there is a true effect of the intervention, which increases scores by 0.3 s.d. units. (i.e. the effect size is 0.3). On the other 50% of trials, the intervention has no effect.*

*Your aim is to earn as many points as possible by identifying which trials show data from a population in which there is a true effect, and which trials come from a population with no effect.*

*The goal is for you to get a sense of how big a sample you need to make this judgement.*

*The game will include 80 trials in total, with breaks in between each block of 20 trials.*

*Every trial will start by showing a few data points from each group.*

*At that stage you can EITHER:*

*judge whether there is a true effect or no effect, by pressing one of the buttons below the data,*

*OR*

*you can wait a few seconds for the screen to automatically advance and show you more evidence (additional data points).*

*Collecting more evidence can make you more sure of your decision.*

***Please note: this study does NOT involve any deception. The feedback you get will be accurate, even if it is sometimes surprising.***

*Each trial will start with an image like this, showing a set of data points: one from the control group and one from the experimental group.*

*Along the y-axis you will see scores for each data point. Along the x-axis you will see how many data points there are in each group. Data points for the control group are pink and data points for the experimental group are blue. Each screen will start with 10 data points per condition.*

*If you wait for more evidence, then more coloured data points will appear.*

*Each new array of data points doubles in size.*

*You can wait for more data until the target sample size of 320 is reached.*

*While data accumulates, you will have to judge whether you think the samples show a true effect or no effect. In other words—do these samples come from populations with different means, or the same mean?*

*If you think these samples came from populations that truly differ, respond with **Blue > Pink**.*

*If you think these samples came from populations that are the same, respond with **Blue = Pink**.*

*Once you answer you will hear a bloop sound indicating that your response has been recorded.*

*After your response is made, the display increases until there are 320 data points in each condition.*

*You will then see a feedback slide telling you whether you answered correctly and displaying how many points you earned.*

*On each trial, you earn 4 points for a correct answer, but you lose 4 points if you make a mistake.*

*You can also earn 2 bonus points if you make a correct response at the optimal array size. This is the smallest array size at which the likelihood of one scenario (true effect or null) is 40 times greater than the other. You should get a sense of what this looks like as you go through the game. The optimal array size will vary from trial to trial, just because of chance factors.*

*You should try to maximize your score as we will convert your points into pennies and add these to your payment. Don't worry, though: if you make lots of errors and end up with a negative score, we won't subtract anything from the basic payment for this session.*

The participant is then shown some examples that demonstrate what the stimuli look like, and the reward schedule is explained.

At the end of training, they are asked if they thought they had got better at the task over time (options: Yes, Unsure or No) and whether they adopted any specific strategy to guide their responses (free text response box).

**Training items**

Gifs for training items were generated by a purpose-designed script which is available on Open Science Framework (https://osf.io/x65nk/?view_only=b0fcb097cc3044aaa942f15136441c49). For each item, samples of observations were selected in a cumulative fashion, such that, for instance, the sample with 20 observations per group included the sample with 10 observations per group, plus an additional 10 observations. The array sizes doubled with each step, corresponding to 10, 20, 40, 80, 160 and 320 observations per group.

All items included a pink sample from a population with mean of zero and s.d. of one, plus a blue sample. For the blue sample, half the items were drawn from the same population as the pink sample (i.e. null effect) and half were drawn from a population with mean of 0.3 and s.d. of one (i.e. true effect). Figure 3 illustrates a set of samples shown side by side for a trial with a true effect. Figures were created using the R programming language [14] using the *geom_beeswarm()* function from the *ggbeeswarm* package (v. 0.6.0) [15] in conjunction with *ggplot2()* (v. 3.2.1) [16]. Each plot was combined into a gif using the *transition_states()* function from the *gganimate* package (v. 1.0.7); [17]. In Pilot 1 (see below), we showed only the data points corresponding to the two datasets, but, as discussed below, for Pilot 2 we added the sample mean shown as a horizontal bar with the aim of improving accuracy.

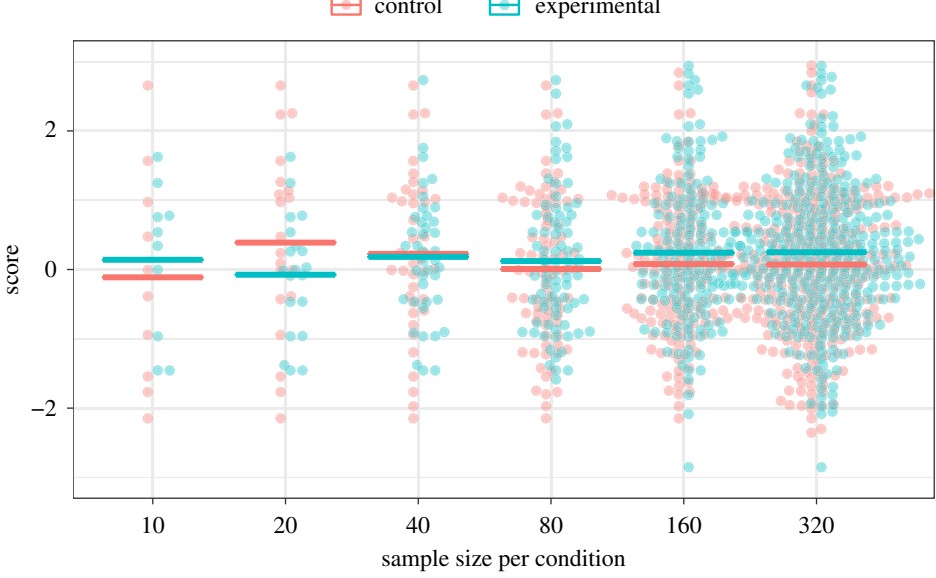

**Figure 3.** Sample item from the training set. The pair of distributions corresponding to each sample size is presented in a gif one at a time sequentially for 2 s, so the display builds up from left to right over time.

Items from each figure were animated so that each pair of pink-blue observations at a given array size was shown for 2 s, before the next array size appeared. With this method, response latencies can be directly converted to the array size at the point of response, i.e. responses under 2 s correspond to array size of 10 per group, those under 4 s to an array size of 20 per group, and so on. For analysis, array is coded as an array index from 1 (10 per sample) to 6 (320 per sample).

Participants received feedback after each response, in the form of a display showing points earned or lost, plus total points so far, and a violin plot showing the full distribution from which the points were drawn (figure 4).

Data from the training trials were saved as response latency, which was converted to array size, as described above. For data analysis, array size (10, 20, 40, 80, 160 and 320) was converted to array index, ranging from 1 to 6. In addition, stored with each array is the amount earned, true effect size in the population on that trial (null or 0.3), coding of each response as correct or incorrect, the observed effect size on that trial, and the log likelihood of a true versus null effect obtained from the observed data. The latter two variables could potentially be used in exploratory analyses when evaluating which cues participants base their responses on.

This basic task evolved over three pilot studies, which led to improvements in the design. A document with details of pilot tasks and results is available on Open Science Framework (https://osf.io/s39qd/). In two of the pilot tasks (1 and 3), no learning occurred, but in Pilot 2, there was clear evidence of learning. Accordingly, we used this version of the task. As well as exploring the influence of changes to the displays viewed by participants, we recognized the importance of manipulating incentives to make participants take the task seriously. In the current version, we adopted the method used in Pilot 2, which was to encourage thoughtful responding by explaining the idea of an optimal array size—which is the smallest array size at which the odds in favour of either the null or true hypothesis reaches $40:1$ (absolute log likelihood of 3.68) or above. We also converted the total positive points earned (4p for correct, −4p for error, plus bonus) into a financial bonus added to the basic payment for participation.

## 3.4. Analysis plan

### 3.4.1. Exclusionary criteria

Our pre-registration stated that if participants did not complete the post-task quiz they would still be included in the main analysis of learning effects, but replacement participants would be recruited to ensure a sufficient sample size for the pre- and post-training analysis of the quiz results. In practice, this was not needed, as all participants completed pre- and post-training quizzes.

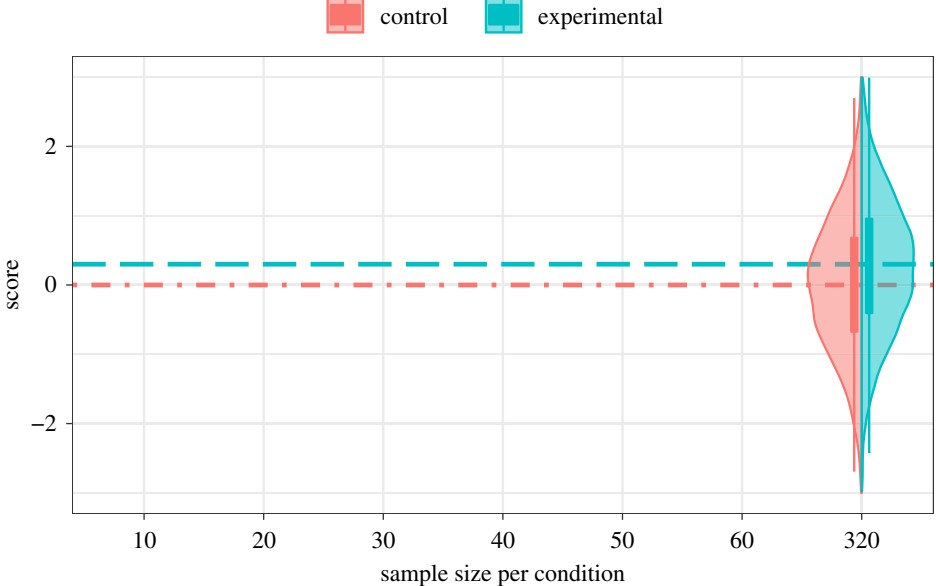

**Figure 4.** Feedback display for an item with a true effect.

We also specified we would exclude from the main analysis any participants who scored 5 or 6 correct on S-items in the pre-training quiz, as they would be people with a good prior understanding of sample size effects, so would be unlikely to show learning gains. This led to exclusion of 2/100 participants.

For the regression analysis, we used an index of learning which corresponds to the difference in earnings between first and last blocks.

## 3.5. Data analysis

Pre-registered analyses were designed to test five predictions that follow from the three-step hypothesis outlined above:

*Prediction 1*: Responses will reflect overconfidence in small samples.

A simple preliminary check was made using a one-tailed $t$-test to check whether the mean number of S-items correct on the pre-training quiz is less than 5/6. If this prediction were not confirmed, then this would undermine hypothesis (a) and hence the rationale of the study. Thus, this test may be regarded as a form of positive control. In addition, a chi-square analysis of error responses was conducted on pre-training S-items to test the expectation that errors on S-items would not be random, but would cluster on responses that indicate either that sample size is not important, or, if asked to select an optimal sample size, involve selecting one that is smaller than required. (These responses are marked x in the Appendix 1).

*Prediction 2*: Accuracy will increase with training.

We anticipated replicating the results of Pilot 2, showing improved performance, as reflected in earnings, with exposure to the beeswarm task. A one-tailed matched pairs $t$-test was used to compare earnings on block 4 with earnings on block 1, with the prediction that the block 4 score would be higher.

*Prediction 3*: Increased accuracy will be associated with selection of larger array sizes as learning proceeds.

This is a subsidiary prediction to Prediction 2, which may throw light on the heuristics used by successful participants, if learning is observed. We predicted that, as found in Pilot 2, there would be a significant association between mean array size and mean earnings, such that those who earn most would be those who selected larger array sizes (tested by the significance of Pearson correlation between these variables in block 4).

*Prediction 4*: Self-reported strategy will be predictive of learning.

Again, this was a subsidiary prediction to Prediction 2, with potential to clarify which cues are used by successful learners. We divided the self-reported responses into those that did ($N = 37$) and did not ($N = 61$) indicate awareness that larger sample sizes (or longer waiting) give more reliable estimates.

This was tested using a one-tailed independent groups *t*-test. Consistent with Pilot 2, we predicted that those who showed such awareness would obtain higher earnings in the final block than those who do not.

*Prediction 5*: Generalization of learning.

Individual differences in learning on the beeswarm task will predict improvement on the estimation quiz, specifically for items that are designed to assess sample size neglect.

The prediction was tested using linear regression:

$$postS \sim preS + earn.diff,$$

where postS and preS are post-training and pre-training scores on S-quiz items, and earn.diff is the difference in earnings (a measure of success on the task) between the last and first block of training. This simple approach to analysis was settled upon after using simulated data to conduct power analysis comparing linear regression versus a linear mixed models approach. The script for simulation is available on OSF (https://osf.io/kcrvw/files/).

Further exploratory analysis were used to scrutinize the response profiles on the beeswarm task for those participants who did show learning, as well as their self-report of strategies, with the aim of determining which cues they relied on when making a response (see Exploratory analyses, below).

The positive control P-items from the quiz were scrutinized to check that the participants were competent in their knowledge of basic probability and motivated to respond accurately. A pre-registered criterion was that if more than half the participants scored less than 3/6 correct on P-items, then this would undermine the interpretation of poor performance on S-items.

## 3.6. Stopping rule

We proposed an initial check after testing 30 participants to see whether Prediction 1 was supported, indicating that participants on average show sample size neglect. This was confirmed. If it had not been, then we would have halted data collection, and not proceeded with the Registered Report, as the basic premise of the study would not be supported. As noted above, Prediction 1 acted as a positive control.

Assuming the study continued, we proposed to stop and check results after testing 50 participants and then after each increment of 25 participants. Because power depends crucially on the proportion showing learning, and the increase in per cent correct with learning, both of which were unknown, we planned to adopt a Bayesian stopping rule at sequential stages in data collection, starting with a sample of 50, and then increasing to 75 and 100 if necessary to meet our pre-specified criteria. We planned to compute Bayes factors for the *t*-test analysis testing Prediction 2, and the linear regression for testing Prediction 5. When Bayes factors yield evidence in favour of the alternative or null hypothesis for both analyses, we would stop data collection.

## 3.7. Exploratory analyses: pilot data

To complement the analysis of self-reported strategies (Prediction 4), the pattern of responses on the beeswarm task was explored to see whether individual participants adopted specific strategies. Accuracy scores were used to compute signal detection indices, d prime and beta, for each block of training, based on the number of hits (respond yes when there is a true effect); the number of misses (respond no when there is a true effect), the number of false alarms (respond yes when there is a null effect) and the number of correct negatives (respond no when there is a null effect). Calculations of these indices was conducted using the *dprime()* function from the *psycho* package (v. 0.5.0) [18]. The beta values allow us to detect cases of response bias, i.e. a tendency to always respond 'blue = pink' or 'blue > pink'.

As shown in the power values in figure 2, with an effect size (Cohen's *d*) of 0.3, participants need to wait for an array of at least 80 per group (index 4) to have a reasonable chance of success (above 50%) for detecting a true effect and would be well advised to wait for the largest array (index 6), which would virtually guarantee success. However, the trial-by-trial variation allows us to do a rather more sensitive analysis, testing how the evidence available on each specific trial is used.

Figure 5 illustrates the logic. The figure shows how evidence accumulates over a series of trials, labelled A–G.

Figure 5 shows the contrast between observed effect sizes, which show more variation at small array indices, and log likelihood, where trials with true and null effects diverge as array index increases.

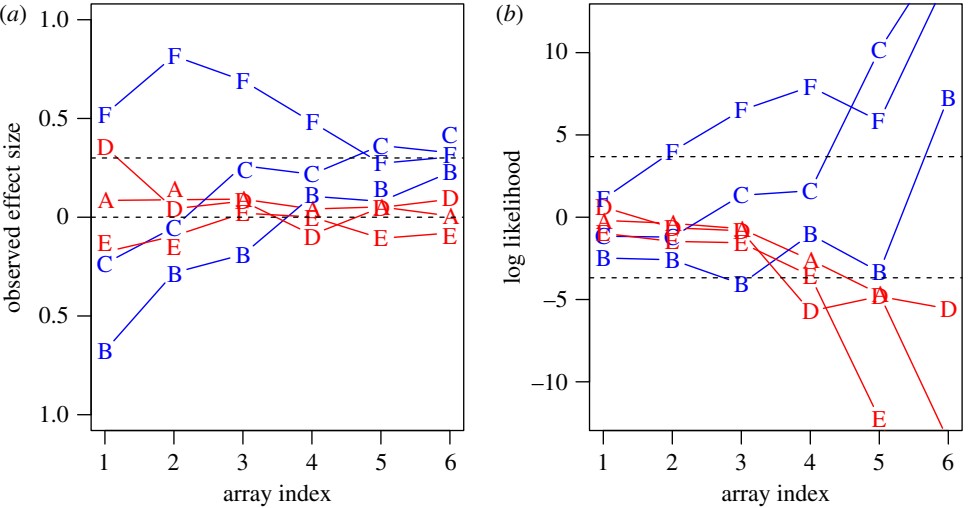

**Figure 5.** Changes in observed effect size (*a*) and log likelihood (*b*) as evidence accumulates for six trials of the learning task, labelled A–G. Blue denotes trials with true effect size of 0.3, and red denotes null trials. Array indices 1 to 6 correspond to sample sizes of 10, 20, 40, 80, 160 and 320 per group.

Considering first the observed effect sizes, it is clear that if a participant relied on this information to make decisions at small array indices, they would make many errors. Two of the true trials, B and C, have negative effect sizes at array indices 1 and 2, and one of the null trials, D, has an effect size greater than 0.3 in the first array. Log likelihood values are shown in figure 5*b*, with dotted lines denoting cut-off of ±3.68, which we used as a criterion for the optimal array size. A log likelihood of 3.68 indicates that the true hypothesis is 40 times more likely to have generated the observed data than the null hypothesis, and a log likelihood of −3.68 indicates that the null hypothesis is 40 times more likely than the true hypothesis. The trials shown in figure 5 indicate that a strategy of responding true when log likelihood exceeds 3.68 and null when it is less than −3.68 is not infallible (e.g. trial B has log likelihood below −3.68 at array index 3, yet comes from a distribution with true effect size of 0.3). Nevertheless, we can use this approach to compute accuracy levels that would arise from adopting different strategies over all trials, and it is clear that responding when absolute log likelihood exceeds 3.68 is a close-to-optimal strategy for achieving success without needing to wait until the final array.

Of course, a problem for participants is that they cannot estimate log likelihood directly from the information they are presented with. However, they can estimate the amount of overlap between the blue and pink samples, which is predictive of log likelihood. Our primary analysis assumes that, on the basis of feedback, participants may simply learn that they are more likely to be accurate if they wait to see more data; in which case, they would base their judgement on observed effect size, once the array index gets to 5 or above. By observing the pattern of responses in individuals, we aimed to determine if a participant adopts a more specific strategy, and if so, whether they placed reliance on observed effect size or sample overlap.

## 3.8. Simulated data for sample size determination

To simulate data, we used our best judgement combined with insights from the pilot testing to explore feasible scenarios. We started by assuming that at the start of training, on each trial, participants will select an array index in proportions similar to that seen in the first block of Pilot 2: index 1 = 0.005, index 2 = 0.022, index 3 = 0.102, index 4 = 0.238, index 5 = 0.358, index 6 = 0.275. By the final block, a proportion (p) of participants will be designated as 'learners'. Whether a participant is a learner is simulated using a variable, *L*, which is latent, in the sense that it affects performance on observed measures, but is not itself detectable. *L* is either 0 or 1. The response made by the participant is determined by three factors: (i) whether the participant is a learner, (ii) whether log likelihood of distributions at that array index favours a true or null effect, and (iii) random error, which leads to a higher proportion of errors in non-learners than learners.

Where *L* is equal to one (i.e. the participant is a learner), the participant waits for an array index where the display corresponds to an absolute log likelihood of at least 3.65. For non-learners, the

**Table 1.** Power for comparison of earnings for block 4 versus block 1, in relation to proportion of learners (plearn).

| parameters | N = 50 | N = 75 | N = 100 |
|---|---|---|---|
| plearn = 0 | 0.04 | 0.04 | 0.04 |
| plearn = 0.25 | 0.68 | 0.77 | 0.81 |
| plearn = 0.33 | 0.88 | 0.94 | 0.97 |
| plearn = 0.50 | 1.00 | 1.00 | 1.00 |

response is made on the basis of whether log likelihood is positive or negative, regardless of the strength of evidence. In addition, a proportion of all responses for both learners and non-learners are coded as errors, with the proportion being higher for non-learners. A simulated sample of participants is created by applying these rules to the stimulus set used in training, with response on each trial being then scored as correct or incorrect.

Prediction 2 focuses on whether there is evidence of learning, with higher earnings on the last block than the first block. The sample size (N subs) was varied from 50, to 75, to 100. The proportion of participants who learn (plearn) could take values of 0, 0.25, 0.33 or 0.5. Table 1 shows the power for detecting an overall change in mean earnings from block 1 to block 4, using a matched pairs one-tailed $t$-test, with alpha of 0.0322 (0.1/3, for one-tailed test administered at each of three sample sizes). A one-tailed test is used because the prediction is directional. These figures were derived from 500 iterations of the simulation.

The case where no participants learn corresponds to a test of the null hypothesis, and so here the power value corresponds to the false positive rate. As expected, this is close to 0.03. At our smallest sample size (50), even when only 25% of participants learn, power is above 0.8, even for the case where learning increases array size by only one index point.

To address Prediction 5, quiz data were simulated using the binomial distribution, assuming the mean number of S-items correct on the pre-test is 2/6 (0.33), improving by either one item (3/6 = 0.5) or two items (4/6 = 0.66) for the subgroup of participants who showed learning on training (where the latent variable, $L = 1$).

Results are shown in table 2, again looking at scenarios with 50, 75 or 100 participants, and where the proportion showing learning ranges between 0, 0.25, 0.33 and 0.5. One other parameter is considered for the quiz data, namely the amount of improvement in quiz scores seen in learners—a mean gain of either one quiz item, or two quiz items.

Power to detect any improvement in the overall sample is generally poor if one just compares quiz pre-test and post-test items, except where the gain in quiz score is at least 2 points and at least 33% of participants learn. This follows because the number of quiz items is small, only a proportion of participants show any learning, and the relationship between learning and quiz post-test performance is imperfect.

An alternative approach uses linear regression, where the focus is on predicting the quiz post-training score (S-items) from the pre-training score and the index of learning, i.e. the change in earnings from block 1 to block 4. This analysis, then, can detect whether the amount of learning in the training session is predictive of improvement on the quiz.

As shown in table 3, the combination of parameters determines power, with high power seen when at least 33% of participants show learning, and when their quiz scores increase by 2 points. Power to detect a 1-point gain in quiz scores for learners is modest at best. Once again, consideration of the case where no participants show learning confirms that the type 1 error rate is well controlled, so we can be confident that where positive findings are obtained, they are true effects.

## 3.9. Interpretation of pattern of results

We can refer to the pattern of results on the beeswarm task and the quiz as indicating null result, inconclusive or prediction confirmed. In our pre-registration, we specified interpretations as follows:

The first possible pattern is where neither the beeswarm task nor the quiz shows any improvement. If this pattern were observed, we would conclude that the beeswarm task was not effective in debiasing participants away from sample size neglect to a meaningful extent. We would consider using information from the experiment to devise a modified version of the task. There are various ways one

**Table 2.** Power for comparison of quiz pre- and post-test, in relation to number of participants (N), proportion of learners (plearn) and increase in quiz score in learners (Quiz gain).

| parameters | N = 50 | N = 75 | N = 100 |
|---|---|---|---|
| Quiz gain = 1 | | | |
| plearn = 0 | 0.02 | 0.03 | 0.03 |
| plearn = 0.25 | 0.14 | 0.18 | 0.24 |
| plearn = 0.33 | 0.22 | 0.32 | 0.43 |
| plearn = 0.50 | 0.45 | 0.64 | 0.8 |
| Quiz gain = 2 | | | |
| plearn = 0 | 0.04 | 0.04 | 0.03 |
| plearn = 0.25 | 0.38 | 0.6 | 0.71 |
| plearn = 0.33 | 0.62 | 0.82 | 0.91 |
| plearn = 0.50 | 0.93 | 0.98 | 1 |

**Table 3.** Power for regression coefficient predicting post-training score on S-items from observed increase in earnings on the learning task. Power varies with number of participants (N), proportion of learners (plearn) and Quiz gain at post-training for learners (Quiz gain).

| parameters | N = 50 | N = 75 | N = 100 |
|---|---|---|---|
| Quiz gain = 1 | | | |
| plearn = 0 | 0.02 | 0.03 | 0.05 |
| plearn = 0.25 | 0.19 | 0.28 | 0.38 |
| plearn = 0.33 | 0.28 | 0.38 | 0.48 |
| plearn = 0.50 | 0.28 | 0.44 | 0.53 |
| Quiz gain = 2 | | | |
| plearn = 0 | 0.04 | 0.03 | 0.03 |
| plearn = 0.25 | 0.57 | 0.73 | 0.86 |
| plearn = 0.33 | 0.68 | 0.84 | 0.95 |
| plearn = 0.50 | 0.77 | 0.89 | 0.98 |

could do this: by changing the visual display, or by changing the reward schedule, or providing more explicit instruction. It would, of course, also be possible to extend the training, although this might require having more than one session. In effect, this study could act as a baseline against which to evaluate other approaches to training.

If results on one or both measures are inconclusive, this could reflect insufficient training/quiz items, and or sample heterogeneity. If the sample is heterogeneous, with a subset responding to the training, we would expect to see a relationship between gains on training and gains on the S-items on the quiz. In addition, we would expect to see an association between learning and self-reported awareness of the importance of sample size. We would also aim in future work to identify characteristics of those who improved with training.

We may find that there are significant improvements with training, but null or inconclusive results on the quiz, indicating a lack of generalization. If so, we would explore individual S-items in the quiz, to see whether there is any indication of improvement on a subset of items, and whether they have particular characteristics. We may, for instance, see evidence of selective improvement on items assessing understanding of the relationship between sample size and statistical power, which could be tested in a replication study.

The converse pattern might be obtained, with improvement on the S-items of the quiz but null or inconclusive findings on the beeswarm task. This might justify a further study which uses two training sessions, making it possible to look for gains due to training that occur after a delay.

Finally, if we are able to show an improvement with this short training session, with generalization to S-items on the quiz, then this would lead us to conclude that the game might be a useful adjunct to statistical training courses for scientists. This could motivate further studies with modified forms of presentation, to identify the optimum conditions for training, and with different effect sizes. It would also be of interest to evaluate the training for longer term impact on subsequent experimental practices of the trainees.

Tests of Predictions 3 and 4 have potential to provide converging evidence on whether sensitivity to array size is a factor determining learning. If neither prediction is confirmed, but the training is effective, this would suggest we should explore alternative heuristics that might be used by participants to support a successful performance.

## 3.10. Pilot data

Three pilot studies were conducted for the Stage 1 Registered Report submission. Summary results for all three pilots are available on OSF (https://osf.io/s39qd). Overall performance on Pilot 1 was poor, with mean of only 65% correct, confirming the difficulty that participants had in judging effect sizes from distributions. A score of 85% or above was observed in only 10% of all blocks x participants. Participants in Pilot 1 found the task reasonably straightforward but scrutiny of individual cases for evidence of response strategies suggested that only a few participants had started to wait longer to respond as the session proceeded, and others went in the opposite direction, selecting smaller arrays in later blocks. In general, high performers were those who waited for large arrays from the outset, rather than those who changed strategy in the course of training.

Pilot 1 participants also completed the estimation quizzes that preceded and followed the training session. As noted above, modifications were made to wording of some quiz items for the current study on the basis of pilot testing to avoid confusing or over-complicated language.

On the basis of Pilot 1 data, we considered ways of making the training more effective. In Pilot 1, the beeswarm displays for the two groups had omitted the horizontal line corresponding to the mean, and each beeswarm faded out as the next one appeared, which meant it was less obvious that the distributions were cumulative. In addition, although the participants were shown a demonstration of the task, they did not have any familiarization trials, and it was felt that they might perform better if given the chance to try a few trials to become familiar with the pace of the task. For Pilot 2, the displays included a horizontal bar corresponding to the mean, and prior displays remained visible rather than fading out, to emphasize that, within a trial, each set of points included the prior, smaller set. In addition, participants were given four practice trials before the training blocks, and the reward schedule was modified to provide stronger motivation to succeed, as described above.

A second pilot study, Pilot 2, was run with 30 participants on the new version of the beeswarm task (without the quiz). This confirmed that not only was performance improved relative to Pilot 1, but also there was evidence of learning in terms of the percentage correct and earnings measures (which were highly intercorrelated). The mean array index was numerically larger for the last versus the first block, but the difference was small and not statistically significant. However, for the final block, there was a significant correlation between mean array index and per cent correct, and several participants reported that they learned to wait for larger arrays in the course of training, suggesting the task was effective in achieving its aim.

At the suggestion of a reviewer, we conducted Pilot 3 with 20 participants, using an unpaced version of the task, in which on each trial, the participant was presented with a menu and required to select a sample size to view. A plot showing all sample sizes, similar to figure 3, was shown alongside feedback on accuracy for each trial. A cost of one point was incurred with each increase in the selected array index, to prevent participants from always selecting the largest array size. No learning was observed in Pilot 3, which gave results similar to Pilot 1, and therefore this approach was abandoned.

# 4. Results

In a departure from our pre-registered plan, we present here results from two samples of participants. Sample 1 ($N = 50$) was recruited as planned, but, owing to a coding error, the estimation quiz items were administered in a non-random order, with the six P-items followed by the six S-items. As we could not be sure of the effect of this procedural change on performance, we recruited sample 2

($N = 50$), who were administered the estimation quiz with items randomized as intended. Preliminary analyses (see below) indicated no meaningful differences on quiz scores between the two samples.

On the basis of the Bayesian stopping rule that we pre-registered, the 50 participants in sample 2 gave only anecdotal evidence of learning on the training game. As there were no differences between the two samples in administration of the training game, we therefore combined samples 1 and 2 to give a total sample of 100 participants for our main analysis.

## 4.1. Demographics

Summary data for participants are shown in table 4. There was a wide spread in terms of self-rated statistical competence (see Appendix 1 for questions) and educational level.

## 4.2. Confirmatory analyses

We tested our five pre-specified predictions based on the pre-registered analysis plan.

*Prediction 1*: Responses on quiz items will reflect overconfidence in small samples.

We first checked the pre-test quiz data against two pre-specified criteria: (i) that mean number of S-items correct is less than 5/6 (83%) and (ii) that fewer than 50% of participants score less than 3/6 correct on P-items (see prediction at end of Data analysis section). Both predictions were confirmed. In sample 1, the pre-training mean on S-items was 1.7, $t$-test of difference of mean from 5 = $-17.17$, $p < 0.001$; in sample 2, the pre-training mean on S-items was 1.96, $t$-test of difference of mean from 5 = $-15.53$, $p < 0.001$. The percentage of participants scoring less than 3/6 on the pre-training P-items was 34% for sample 1 and 36% for sample 2. We also considered which was the most common response for each item on initial testing: for all 12 P-items, the correct response was the most common, whereas this was the case for only 5/12 S-items. We found that two participants scored 5 or more on S-items correct prior to training, and they were dropped from the learning analyses, consistent with our pre-registration.

Next we checked if the two samples differed in their responses to the quiz items, by running a linear mixed model predicting proportion correct from item type, pre-/post-testing and sample, with the formula:

$$\text{lmer(p.correct} \sim \text{itemtype} + \text{prepost} + \text{sample} + (1|\text{subject}),$$

where itemtype was whether P- or S-item, pre-/post-specified whether the quiz was before or after training, sample was sample 1 or 2, and subject was a random effect. This model was compared with a model where the sample term was omitted, to give a Bayes factor of 0.008, giving strong evidence for the null hypothesis of no difference between samples. As shown in table 5, the analysis also indicated a substantial effect of item type, reflecting the superior performance on P-items, but no effect of whether the quiz was administered pre- or post-training. We show data separately for the two samples in some subsequent analyses, but they were treated together for the main analysis.

Table 6 shows errors on the pre-training quiz categorized as to whether they reflected sample size neglect (bias) or not (Other1 and Other2). The 'bias' foil in each case explicitly mentioned that sample size (or gathering more data) would have no effect. Items S3A and S3B are omitted from this table, as there is no specific foil that corresponds to this bias. For each item, the distribution of error responses was subjected to chi-square test, to test the hypothesis that all foils were equally likely to be selected. For most items, the chi-square value was statistically significant, indicating non-random choice of foils, and for items 1A, 1B, 2B, 5A and 6B the most common choice was the foil designated as reflecting sample size neglect. Items 1 and 5 were based on two of Tversky and Kahneman's classic examples of 'law of small numbers' (probability 60% boys being born in a big or small hospital, and likelihood of best player winning increasing with number of games); item 6 focused on how type II error rate is influenced by sample size. On items 2A, 4A and 4B, there was strong bias to select another foil. We discuss response patterns to specific items in Exploratory analysis below.

*Prediction 2*: Accuracy on the beeswarm task will increase with training.

Figure 6 shows the distribution of earnings for the first and last blocks in the two samples. A matched pairs $t$-test was used to compare earnings on block 4 with earnings on block 1, using a one-tailed test, as the prediction is that the block 4 score will be higher. This confirmed that learning had occurred in both samples: sample 1: $t_{49} = 4.03$, $p < 0.001$; sample 2: $t_{47} = 1.8$, $p = 0.039$. However, for sample 2, the Bayes factor (0.711) provided only anecdotal evidence against the null hypothesis, whereas for Sample 1, the evidence was solidly in favour of the alternative hypothesis (that the improvement in earnings was

**Table 4.** Demographics and self-rated statistical skills of participants. Notes: *'Compared to others at your level of education in your participant, how good do you think your understanding of basic statistics is?'; rated on scale from 0 (very poor) to 100 (excellent). **'How confident are you in interpreting the results of a *t*-test?' rated from 1 = very confident to 4 = Very little idea of what a *t*-test is. ***'How familiar are you with the idea of statistical power?' rated from 1 = very familiar to 4 = very little idea about what statistical power is'.

| | sample 1 (N = 50) | sample 2 (N = 50) |
|---|---|---|
| sex | | |
| female | 34 (68.0%) | 40 (80.0%) |
| male | 16 (32.0%) | 9 (18.0%) |
| other | 0 (0%) | 1 (2.0%) |
| age band | | |
| 18–24 | 27 (54.0%) | 34 (68.0%) |
| 25–30 | 11 (22.0%) | 6 (12.0%) |
| 31–40 | 10 (20.0%) | 6 (12.0%) |
| 41–50 | 2 (4.0%) | 4 (8.0%) |
| Educational level | | |
| pre-degree | 5 (10.0%) | 3 (6.0%) |
| 1st degree | 33 (66.0%) | 28 (56.0%) |
| masters | 7 (14.0%) | 15 (30.0%) |
| doctorate | 5 (10.0%) | 4 (8.0%) |
| Statistics self-rating* | | |
| mean (s.d.) | 54.8 (19.2) | 50.2 (16.3) |
| median [min, max] | 57.5 [15.0, 99.0] | 50.0 [12.0, 85.0] |
| Understand *t*-test** | | |
| mean (s.d.) | 2.54 (0.885) | 2.44 (0.705) |
| median [min, max] | 2.00 [1.00, 4.00] | 2.00 [1.00, 4.00] |
| Understand power*** | | |
| mean (s.d.) | 2.74 (0.876) | 2.66 (0.895) |
| median [min, max] | 3.00 [1.00, 4.00] | 3.00 [1.00, 4.00] |

**Table 5.** Estimates of effects of item type (P- versus S-item), pre-/post-administration and sample on quiz accuracy.

| source | estimate | s.e. | *t*-value |
|---|---|---|---|
| (intercept) | 0.575 | 0.028 | 20.357 |
| item type (P/S) | −0.276 | 0.019 | −14.208 |
| pre/post | 0.015 | 0.019 | 0.789 |
| sample (1/2) | −0.037 | 0.035 | −1.049 |

greater than zero), Bayes factor = 180.701. With data from both samples combined, the Bayes factor was 164.222, giving overwhelming evidence of learning.

*Prediction 3*: Increased accuracy will be associated with the selection of larger array sizes as learning proceeds.

We predicted a significant association between mean array index and mean earnings, such that those who earn most will be those who select larger array sizes (tested by the significance of Pearson correlation between these variables in block 4).

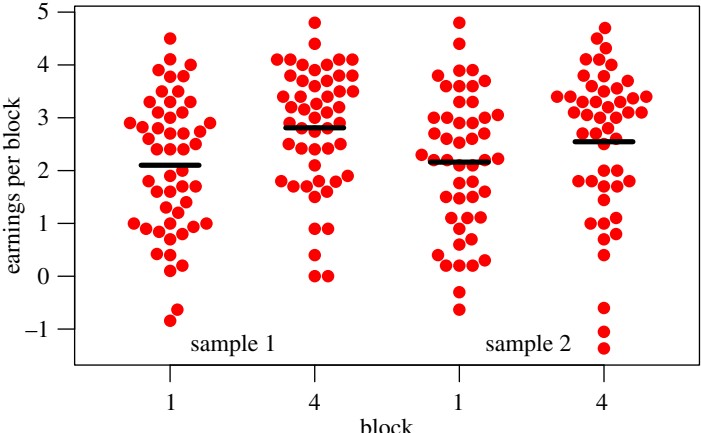

**Figure 6.** Distribution of earnings in first and last blocks for samples 1 and 2.

**Table 6.** Distribution of error responses on quiz pre-training S-items.

| item | bias[a] | Other1[a] | Other2[a] | chi-sq | $p$ |
|------|------|--------|--------|--------|-----|
| S1A | 28 | 7 | 4 | 26.31 | 0.000 |
| S1B | 22 | 8 | 2 | 19.75 | 0.000 |
| S2A | 9 | 1 | 34 | 40.41 | 0.000 |
| S2B | 12 | 4 | 2 | 9.33 | 0.009 |
| S4A | 12 | 29 | 3 | 23.77 | 0.000 |
| S4B | 8 | 11 | 29 | 16.12 | 0.000 |
| S5A | 21 | 9 | 6 | 10.50 | 0.005 |
| S5B | 10 | 15 | 8 | 2.36 | 0.307 |
| S6A | 14 | 14 | 4 | 6.25 | 0.044 |
| S6B | 23 | 7 | 2 | 22.56 | 0.000 |

[a]Note: Columns show $N$ selections of each foil (samples 1 and 2 combined), where bias indicates a foil that demonstrates sample size neglect, and Other1 and Other2 denote the other two foils in the order they were shown. The Chi-square value tests whether, for items answered incorrectly, the selection of foils is random.

As shown in figure 7, this prediction was confirmed in both samples. For the combined sample, $r_{96} = 0.472$, $p < 0.001$.

*Prediction 4*: Self-reported strategy will be predictive of learning.

We divided the self-reported responses into those that did and did not indicate awareness that larger sample sizes (or longer waiting) give more reliable estimates. See online Appendix 2 (https://osf.io/uej95/) for response coding by two of the authors—this agreed for all but one participant in each sample; analysis was based on coding by J.T. Consistent with Pilot 2, we confirmed the prediction that there would be higher earnings in the final block for those who showed such awareness: one-tailed *t*-test in sample 1: $t_{47.2} = -2.25$, $p = 0.029$, and in sample 2: $t_{42.1} = -2.55$, $p = 0.014$, and in the combined sample $t_{95.5} = -3.48$, $p < 0.001$.

*Prediction 5*: Generalization of learning.

As noted above, there was no indication of improvement on the post-training estimation quiz in either sample, making it unlikely that the pre-registered analysis would find any generalization of learning gains on the beeswarm task to the S-items on the quiz.

The results of the linear regression (table 7) confirmed this was the case: individual differences in learning on the beeswarm task did not predict improvement on the S-items of the estimation quiz. The model was specified as

$$postS \sim preS + earn.diff,$$

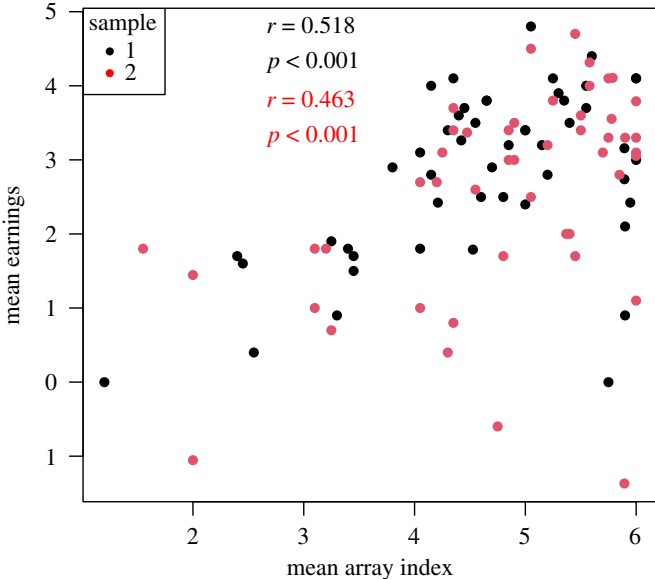

**Figure 7.** Association between array index and earnings in samples 1 and 2.

**Table 7.** Linear regression results for predicting per cent correct on S-items on post-training quiz, from pre-training score and learning index (earnings difference from first to last block).

| source | estimate | s.e. | *t*-value |
|---|---|---|---|
| intercept | 0.251 | 0.036 | 6.981 |
| pre-training S-items correct | 0.112 | 0.092 | 1.208 |
| learning index | 0.001 | 0.015 | 0.080 |

where postS and preS are post-training and pre-training scores on S-items, and earn.diff is the difference in earnings (a measure of success on the task) between the last and first block of training.

The learning index from the beeswarm task did not predict the post-test score on the S-items from the quiz, and the Bayes factor (0.1), computed by comparing the Bayesian information criterion (BIC) of the full model, with the BIC from the model omitting the learning index, provided moderate support for the null hypothesis.

### 4.3. Exploratory analyses: quiz data

Given the overall lack of improvement on quiz items, we did not conduct any further analysis to see if specific items improved, but we did look more closely at the errors made on S-items, in an attempt to understand what was driving responses.

Items 1A and 1B were somewhat analogous to the 'proportion of male births' item from the original study by Tversky & Kahneman [1], although we focused on means rather than proportions, to make the question more relevant to the training. The most common response (both before and after training) was to select a foil that stated that sample size did not matter, even though in training, the participants had been able to see that the mean bars for small samples were far more variable than those of large samples. These items, then, confirmed sample size neglect is commonplace, but that the training with beeswarms showing different sample sizes did not lead to insight on the quiz.

Items 2A and 2B were both designed to test understanding of the fact that a result in the 'wrong direction' was more common with small than large samples, but the two items proved to be of very different level of difficulty. For item 2A, the majority of participants concluded that a small sample ($N = 10$) with mean 158 was equally likely to have come from population A (mean 160, s.d. 6) or B (mean 158, s.d. 6), rather than the correct answer, which is that B is more likely, but a further 90

samples should be collected. Item 2B was analogous to item 2A, but the response options were rather different: there was no foil in which the two options (been poisoned or not been poisoned) were presented as equally likely. The most common response was the correct answer, which indicated that the observed result was inconclusive and more data was needed.

We discuss items 4A and 4B next, because they were identical to items 2A and 2B in wording, except that the specified sample size was adequate for detecting the effect of interest. Note that because of counterbalancing, each participant saw only one version of each question (kangaroos or archaeology). Error rates on these items were higher than for any other items, with a pronounced preference for a specific foil. For item 4A, where a statistical test would indicate overwhelming evidence for one conclusion (animals poisoned), the most common response was that 100 more animals needed to be tested. This was similar to the response to the analogous item 2B, but whereas this response was correct for 2B (where the sample size was 10), it was not appropriate in item 4A (where sample size was 160).

For item 4B, the preferred response was the same as for the analogous item 2A—that the sample was equally likely to have come from either population—even though in item 2A there were two samples each of 10, and in item 4B, there were two samples each of 160. This preference was equally pronounced after training on the beeswarm task, even though that task had given repeated exposure to samples of 160 per group with effect size of 0.3, confirming that an effect of this size is strong evidence that the sample came from the group with the lower population mean.

Items S3A and S3B relate to statistical power by asking directly about sample size needed to be confident in an effect. Of course, the correct answer will depend on the interpretation of 'confident': in selecting possible responses, we took this to correspond to power of 95% or higher, so that the correct responses would be to select groups of 300 for item 3A (effect size 0.3) and groups of 100 for item 3B (effect size 0.4). For both of these problems, there was a spread of responses, but the most common response was the one we had designated as correct, selected by 32–38% of respondents. Experience with the beeswarm task, however, did not affect responses: the distribution of choices was closely similar before and after training.

Items 5A and 5B were modelled on the squash example of Kahneman & Tversky [19]: 'a game of squash can be played either to 9 or to 15 points. Holding all other rules of the game constant, if A is a better player than B, which scoring system will give A a better chance of winning?'. Item 5A simply expanded the available choices to 3 points, 9 points or 21 points, as well as offering the option that the number of points would make no difference. The most common response on pre-training was 'makes no difference', consistent with sample size neglect. The pattern changed after training, with the correct response (21 points) being selected most often. However, this seems likely to be a fluke rather than indicating new insight in participants, because for item 5B, an analogous item framed around chess, the opposite trend was seen: an initial tendency to select the correct response, with the 'makes no difference' response being preferred post-training. The chess problem had had the logic switched so the task was to judge which scenario gave the weaker player the better chance of winning.

Items 6A and 6B focused on risk of type II error with small samples, i.e. a scenario is envisaged where a large difference is seen even though it is known there is no true effect. In both pre-training and post-training data, there was a fairly equal split between correct responses, indicating that this kind of result is more likely in a small sample, and responses indicative of sample size neglect—with an explicit statement that there will be no effect of sample size.

We defer further interpretation of errors on the quiz to the Discussion.

## 4.4. Beeswarm task

Exploratory analysis on the beeswarm task focused on visual comparisons between those who were categorized according to self-report as being aware that sample size was important (Aware group), versus those who were not (figure 8). These two groups were compared for each block in terms of mean array size at point of response, amount of evidence in favour of one hypothesis at the point of response (mean absolute log likelihood), accuracy (mean d prime) and observable difference between means (mean observed effect size at the point of response). The first point to note is that the data on array size validates the self-rating of strategy, with the Aware group showing an increase in array size over time that is not seen in the Unaware group. On the d-prime measure, the Unaware group does show some evidence of increasing accuracy over blocks, but they lag behind the Aware group. The Aware group show an increase in mean absolute log likelihood over blocks, whereas the Unaware

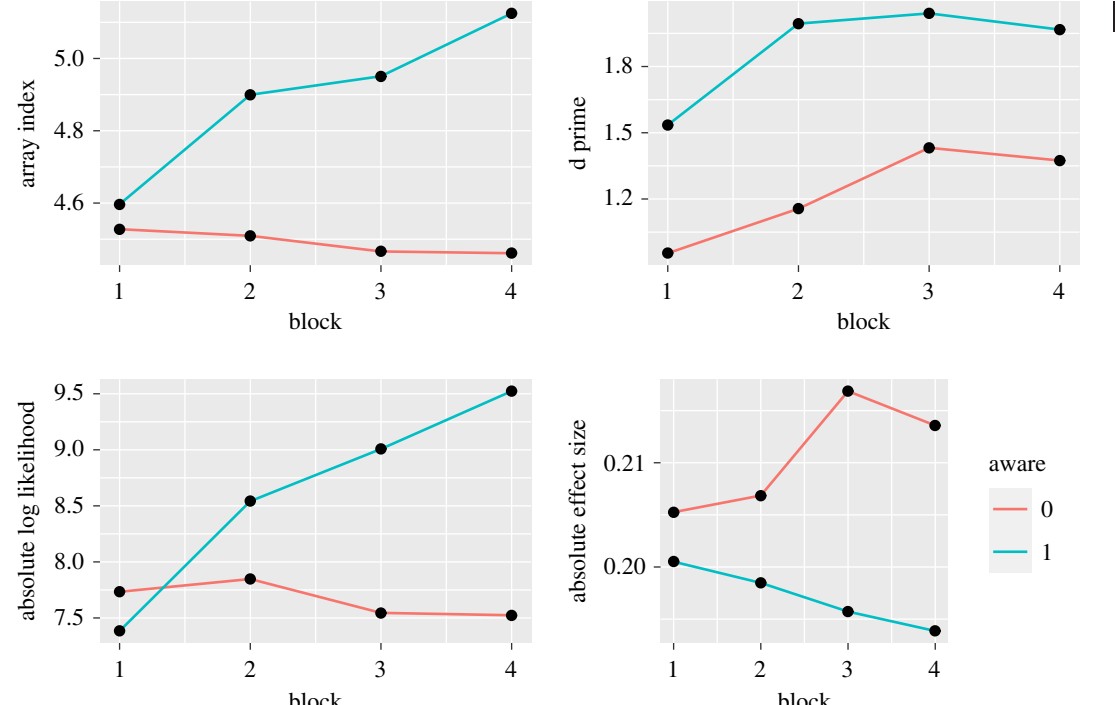

**Figure 8.** Comparison of Aware and Unaware subgroups over four learning blocks, from L to R: mean array size at point of response, amount of evidence in favour of one hypothesis at the point of response (mean absolute log likelihood), accuracy (mean d prime) and observable difference between means (mean observed effect size at the point of response).

group, by contrast, appear to respond more to the observed effect size, i.e. they are influenced more by the difference between blue and pink lines depicting means, ignoring the sample size or variability around the mean. These trends are not surprising, as absolute log likelihood increases and absolute observed effect size decreases with array size.

The substantial differences in learning between participants prompts the question of whether there were any pre-existing differences that might explain why some improved in the Beeswarm game and others did not. Table 8 shows mean scores on self-rating of statistical knowledge and on quiz items, and the 95% confidence interval for the mean difference between the Aware and Unaware subgroups. The Aware subgroup appeared to have better statistical knowledge prior to training, both in terms of self-report, and in terms of performance on the quiz. Nevertheless, there was no indication that their quiz performance improved more with training than the Unaware group.

# 5. Discussion

Results may be summed up as showing: (i) participants were poor at making probabilistic judgements where sample size was critical, (ii) they showed significant improvements on the beeswarm task over sessions, (iii) this was most pronounced in those who were aware of the need to wait for larger samples before responding, and (iv) improvement on the beeswarm task did not generalize to performance on the quiz: scores on parallel forms were similar before and after training.

We will first discuss responses to the quiz, then performance on the beeswarm task, and then the lack of generalization of learning from the beeswarm task. Results on both tasks should be interpreted bearing in mind that participants received monetary bonuses for correct responses on both the quiz and the beeswarm task, and this appeared to be successful in providing motivation to take the tasks seriously. Also, note that our method deliberately avoided mentioning statistical significance or *p*-values at any point in the study, as these can be sources of confusion.

## 5.1. Responses to the quiz

Results confirmed that people were considerably worse at evaluating statements where the correct answer depended on sample size than they were at judging statements that required simple

**Table 8.** Characteristics of participants who were aware or unaware of importance of sample size.

| | unaware (N = 61) | aware (N = 37) | 95% CI for d |
|---|---|---|---|
| **Educational level** | | | |
| pre-degree | 2 (3.3%) | 6 (16.2%) | |
| 1st degree | 43 (70.5%) | 18 (48.6%) | |
| masters | 12 (19.7%) | 9 (24.3%) | |
| doctorate | 4 (6.6%) | 4 (10.8%) | |
| **Statistics self-rating (self-report: 0–100)** | | | |
| mean (s.d.) | 49.9 (17.9) | 56.6 (17.7) | −14.15, 0.61 |
| **Understand *t*-test (self-report: 1 = high, 4 = low)** | | | |
| mean (s.d.) | 2.66 (0.834) | 2.24 (0.683) | 0.1, 0.72 |
| **Understand power (self-report: 1 = high, 4 = low)** | | | |
| mean (s.d.) | 2.90 (0.870) | 2.41 (0.832) | 0.15, 0.85 |
| **Quiz: pre-training P-items correct** | | | |
| mean (s.d.) | 2.87 (1.49) | 3.92 (1.71) | −1.73, −0.37 |
| **Quiz: post-training P-items correct** | | | |
| mean (s.d.) | 3.15 (1.57) | 4.11 (1.54) | −1.6, −0.32 |
| **Quiz: pre-training S-items correct** | | | |
| mean (s.d.) | 1.46 (1.12) | 2.27 (1.45) | −1.37, −0.26 |
| **Quiz: post-training S-items correct** | | | |
| mean (s.d.) | 1.59 (1.07) | 1.89 (1.26) | −0.8, 0.2 |

interpretation or manipulation of probabilities. Furthermore, on items that were closely modelled on those of Kahneman & Tversky [20], response options that stated sample size was immaterial tended to be preferred over the correct option. To that extent, the results support the original work on the 'Law of Small Numbers'. Nevertheless, results varied considerably across quiz items, and in some cases, there was no preference for foils that were selected to detect sample size neglect. Indeed, for some items, participants preferred responses that stated more data should be gathered, even when this was not necessary, suggesting some awareness that sample size could be important.

A particularly intriguing pattern of results was seen on items 2 and 4, which had identical wording, except that on item 2 the sample size was inadequate to demonstrate an effect, and in item 4 it was adequate. At first glance, these items seem to contradict the idea of sample size neglect, because few participants selected responses that explicitly stated that sample size did not matter, and indeed, they tended to prefer responses indicating more data should be gathered. However, their tendency to respond this way was the same regardless of whether they were told that sample size was 10 or 160. Furthermore, for the items where they had the opportunity to say that the sample was equally likely to have come from either group, they preferred that option, regardless of sample size.

With hindsight, it would have been helpful to ask participants to explain the reasoning behind their responses, as this could have thrown light on the unexpected patterns on these two items. Also, the results on items 2 and 4 suggest there could be interest in following the approach adopted in one study by Kahneman & Tversky [20], where the sample size specified in a problem was parametrically varied.

## 5.2. Learning on the beeswarm task

Around one-third of participants indicated that they had learned to wait for larger arrays before responding, and this tendency was associated with higher levels of accuracy on the beeswarm task.

Those who learned to wait for larger arrays showed an increase in accuracy over sessions, whereas those who did not appeared to rely on observed effect sizes, which, in small samples, were an unreliable cue to whether samples came from different populations.

## 5.3. Failure of learning to generalize

The failure to show any improvement on quiz items after training is not encouraging but perhaps should not surprise us. The training was relatively brief, and the link between the insight—that sample size affected the ability to detect a true difference between groups—and the content of quiz items was not made obvious. As Hodgson & Burke [21] have noted, exposure to simulated data alone is not sufficient to inculcate learning: one needs to ensure students are focused on the relevant aspect of a task, with debriefing and follow-up exercises to deepen understanding. Aberson et al. [22] showed improved understanding of the central limit theorem after training on a web-based tutorial where students drew samples of different sizes from a population, but they provided relevant instruction and discussed the concept of the standard error of the mean in depth as part of the training. Other studies have found mixed results from use of simulation in training [23,24], and where learning is found it can remain specific to the precise context of training. In an early study [25], people were given prolonged interactive training to promote understanding of the sampling distribution of the mean, but although this gave better understanding of the effects of sample size, it did not improve ability to answer questions about variability of the mean in different-sized samples. To counteract sample size neglect, we are likely to need more explicit instruction across a range of contexts: for instance, after asking participants to respond to a quiz item, they could be asked to simulate data based on that specific problem, to see how variation in sample size affects the parameter of interest (cf. [26]). Also, consistent with previous suggestions [21,25], we recommend that participants be asked about the reasoning behind their responses to such questions, as this can give insights about the heuristics they adopt.

Kahneman & Tversky [20] regarded the phenomenon of sample size neglect as an instance of the representativeness heuristic, whereby judgements are based on the perceived similarity of a sample to its parent population, and the extent to which the sample reflects salient features of the process by which it is generated. The difficulty in overcoming this bias suggests that representativeness alone may be insufficient to account for it. Rather, it may be that there is conflict between knowledge about statistical parameters, such as mean and s.d., which are unaffected by sample size, and the variability of means, which depends on sample size. Understanding how a mean is computed is an elementary part of statistical training; understanding the nature of variability in that mean comes later in training, if at all, and requires more computational steps. Furthermore, Well et al. [25] found that, when given problems that required answers about the variation in a mean, many participants thought they were being asked about variability in the sample.

Despite decades of research on sample size neglect, it remains a pervasive problem. Given that statistical methods depend on understanding sampling theory, we need to do more to optimize and evaluate methods for developing understanding of the distinction between variation in individual data points and variation in means derived from those data points. Using simulations to give students active experience of characteristics of different-sized samples may be one component in the quest for better methods, but it clearly is not sufficient on its own.

Ethics. The protocol has been approved by the University of Oxford's Medical Sciences Interdivisional Research Ethics Committee, approval number (R60658/RE001).

Data accessibility. This manuscript received Stage 1 in-principle acceptance (IPA) on 12 August 2021, prior to data collection and analysis. The approved Stage 1 version of this manuscript, unchanged from the point of IPA, is pre-registered at https://osf.io/fermv. As stated in the text of the manuscript, data are deposited on Open Science Framework in the following project: https://osf.io/bcpjy/—please note this refers to the RMarkdown manuscript component. The Top Level Component is available at https://osf.io/agn6z/.

Authors' contributions. D.V.M.B.: conceptualization, data curation, formal analysis, funding acquisition, investigation, methodology, project administration, resources, visualization, writing—original draft and writing—review and editing; J.T.: conceptualization, investigation, methodology and writing—review and editing; A.J.P.: conceptualization, data curation, formal analysis, investigation, methodology, project administration, visualization, writing—original draft and writing—review and editing

All authors gave final approval for publication and agreed to be held accountable for the work performed therein.

Competing interests. We declare we have no competing interests.

Funding. This work was supported by Wellcome Trust Programme Grant no. 082498/Z/07/Z.

# Appendix 1: Questions given before and after training

## Initial queries (Pre-training only)

Numbers in square brackets show the frequency of endorsement for each option.

(i) Compared to others at your level of education in your subject, how good do you think your understanding of basic statistics is?
[This item was rated on scale from 0 to 100. Divided into 4 levels at 25,50 and 75]

[10] (a) Excellent
[43] (b) Better than average
[40] (c) Average
[7] (d) Below average

(ii) How confident are you in interpreting the results of a *t*-test?

[12] (a) Very confident
[32] (b) Fairly confident
[49] (c) Not confident
[7] (d) Very little idea about what a *t*-test is

(iii) How familiar are you with the idea of statistical power?
[20] (a) Very familiar
[38] (b) Reasonably familiar
[34] (c) Somewhat familiar
[8] (d) Very little idea about what statistical power is

## Qualitative report (post-training only)

Did you feel you got better at the task over time?
[81] Yes [12] Unsure [7] No

Did you change how you approached the task? Please let us know if you adopted any specific strategy to guide your response?
[See online Appendix 2 (https://osf.io/uej95/) for responses]

## Judgement and reasoning quiz

Participants received version A or B (counterbalanced) at pre-test and post-test. P-items test understanding of probability, S-items test sample size neglect. Items with the same number in version A and B are intended to be analogous but use different contexts. The brief descriptor accompanying each item below is not presented as part of the quiz. An asterisk denotes the correct response. For S-items, x denotes the anticipated response(s) for those susceptible to sample size neglect.

P-items and S-items were inadvertently presented blocked for sample 1, but were randomly intermixed as intended for sample 2. As described in Results, this did not make any difference. This section of the Appendix has now been updated to show the frequency of endorsement of each option in square brackets, first for pre-training and then for post-training administration of the quiz.

### P-items

#### P1a (tests elementary knowledge of normal distribution)

You take a sample of 100 men from the general population and measure their height. The mean is 70 inches and the standard deviation is 4 inches. What percentage of the sample will be expected to be more than 74 inches tall?

[19/22] (a) * 16%
[7/9] (b) 2%
[8/4] (c) 50%
[16/15] (d) 30%

#### P1b

A reading test is standardized on 7-year-old children in Scotland. The mean reading age is 84 months with s.d. of 6 months. What percentage of 7-year-olds is expected to have a reading age of 72 months or less?

[10/17] (a) 16%
[26/11] (b) * 2%
[4/5] (c) 50%
[10/17] (d) 30%

## P2a (basic probability)

A container is full of spare change containing 100 10p coins, 100 5p coins, 200 2p coins and 100 1p coins. The coins are randomly mixed. You will get a prize every time you pick a 5p coin. If you make 100 selections, replacing the selected coin and shaking the jar each time, how often will you expect to pick a 5p coin?

[2/0] (a) One in two occasions
[3/4] (b) One in three occasions
[14/10] (c) One in four occasions
[31/36] (d) * One in five occasions

## P2b

You are choosing marbles from an opaque jar containing 100 red marbles, 100 blue marbles and 200 white marbles, randomly mixed. You will get a prize every time you pick a marble that is not white. If you make 100 selections, replacing the selected marble and shaking the jar each time, how often will you expect to get a prize?

[31/30] (a) * One in two occasions
[7/7] (b) One in three occasions
[10/11] (c) One in four occasions
[2/2] (d) One in five occasions

## P3a (classic probability including averaging probabilities)

In the city of Ficticium, there is generally a 10% chance it will rain on any given day in the first half of September, a 50% chance it will rain any given day in the second half of September, a 25% chance it will rain on any given day in November, and a 30% chance it will rain on any given day in April. (September, November and April are all 30 days long). Which month is likely to have more rainy days?

[15/15] (a) September
[2/2] (b) November
[14/5] (c) April
[19/28] (d) * September and April are equally likely

## P3b

After reviewing sales of coffee, the barista noted there is generally a 50% chance of selling an espresso on any day in the first week of the month. There is a 100% chance of selling an espresso on any day in the second week of the month, and a 75% chance of selling an espresso on any day in the third and fourth week of the month. In what two-week period in the month is there most sales of espresso?

[28/30] (a) * First two weeks and second two weeks are equally likely
[10/0] (b) First two weeks
[11/20] (c) Second two weeks
[1/0] (d) First and fourth weeks

## P4a (basic probability)

At a raffle, you can pick from a green bowl with two winning raffle tickets and eight worthless tickets, a yellow bowl with 10 winning tickets and 90 worthless tickets, or a red bowl with 15 winning tickets and 85 worthless tickets. Which bowl gives you a better chance of winning?

[31/37] (a) * Green
[5/1] (b) Yellow
[12/9] (c) Red
[2/3] (d) Green and Yellow give the same chance

### P4b

You find three bags of jelly beans. There is a small bag with five red jelly beans and 20 other-coloured beans. There is a medium bag with 12 red jelly beans and 38 other-coloured beans. Finally, there is a large bag with 18 red jelly beans and 82 other-coloured beans. As red jelly beans are your favourite flavour, which size bag gives you the highest percentage of red beans?

[13/10] (a) Small
[32/36] (b) * Medium
[4/3] (c) Large
[1/1] (d) They are all the same

### P5a (conjoint probability)

There are 100 girls in a class, 20% have red hair and 40% are taller than 60 inches. How many red-headed girls would you expect who are taller than 60 inches?

[5/2] (a) 60
[7/7] (b) 16
[30/40] (c) * 8
[8/1] (d) 5

### P5b

In a box of 100 CDs, 40% of the disks have been recorded by pop artists and 60% of disks feature female vocalists. Assuming that the female vocalists are equally likely to be pop artists or not, how many CDs have been recorded by female pop vocalists?

[4/1] (a) 12 CDs
[7/2] (b) 18 CDs
[33/28] (c) * 24 CDs
[6/19] (d) 30 CDs

### P6a (classic probability—multiplication)

Ten per cent of the children in a school have red hair. Their names are put in a hat and you are asked to pull out two of them. What is the probability that you will select two red-headed children?

[11/7] (a) 20 per cent
[26/33] (b) * 1 per cent
[1/4] (c) 19 per cent
[12/6] (d) Close to zero

### P6b

Twenty per cent of the 500 children in a school have a name beginning with J. Their names are put in a hat and you are asked to pull out two of them. What is the probability that you will select two children whose name begins with J?

[25/24] (a) * 4 per cent
[10/3] (b) 1 per cent
[6/0] (c) 40 per cent
[9/23] (d) 20 per cent

## S-items

### S1a (frequency of extreme scores in small samples)

History courses in the UK are rated on a Student Survey, which has an overall satisfaction rating of 1 (low) to 5 (high). The mean rating overall is 3.5, with s.d. of 1. Those who have an average rating above 4.0 are given a gold star in league tables. There are 60 courses altogether, which can be divided according to size into small (less than 10 students), medium (between 10 and 50 students) and large (50 + students). If there are no real differences in student satisfaction, will the number of students affect the likelihood of getting a gold star?

[7/9] (a) Yes. Large courses have a better chance of getting a gold star
[4/1] (b) Yes. Medium-sized courses have a better chance of getting a gold star
[11/14] (c) * Yes. Small courses have a better chance of getting a gold star
[28/26] (d) x No. The number of students makes no difference

## S1b

A task force is looking at characteristics of schools in its area. They take a measure of mathematical ability and identify schools as 'failing schools' where 20% or more pupils get scores more than 1 s.d. below the population average. The smallest schools have on average 100 pupils, middle-sized schools have 250 pupils and the largest schools have 500 pupils. If there are no real differences between schools, will the size of the school affect whether it is a failing school?

[18/16] (a) * Yes. The smallest schools will be more likely to be a failing school

[22/27] (b) x No. All else being equal, school size should make no difference

[8/7] (c) Yes. The largest schools will be more likely to be a failing school

[2/0] (d) Yes. The middle-sized schools will be more likely to be a failing school

## S2a (probability of result in 'wrong' direction is higher with small sample size)

A forensic archaeologist has a set of 10 male skeletons from an ancient burial site. Experts are divided as to whether the site was colonized by Tribe A or Tribe B, the only two tribes in the region. Previous studies with large samples found that men from Tribe A had an average height of 160 cm, with s.d. of 6, and men from Tribe B had an average height of 158 cm, with s.d. of 6.

The mean height of the sample is 158 cm. What can the archaeologist conclude?

[1/1] (a) We can be very confident that the sample comes from Tribe B

[6/6] (b) * Tribe B is more likely than Tribe A, but we should collect 90 more samples to be sure

[34/33] (c) Equally likely the sample comes from Tribe A or Tribe B

[9/10] (d) x Can't be sure: Tribe B is more likely than Tribe A, but collecting more samples won't help

## S2b (probability of result in 'wrong' direction is higher with small sample size)

There are concerns that kangaroos appear unhealthy in a particular area of the Australian bush. There is concern that this may be due to eating poisoned bait that affects the blood. A naturalist has blood samples from 10 kangaroos and measures a distinctive blood marker that is lowered in poisoned animals. Previous studies have found that the mean blood marker in healthy kangaroos is 130 with s.d. of 30, whereas the mean is 120 with s.d. of 30 in poisoned animals.

The mean blood marker in the sample is 120. Assuming that there is no other explanation than poisoning for an abnormal blood marker, what can the scientist conclude?

[4/5] (a) The kangaroos have definitely been poisoned

[12/12] (b) x The kangaroos probably have been poisoned but can't be sure. Collecting data from more kangaroos won't help

[32/32] (c) * The kangaroos probably have been poisoned, but we would need to collect blood from 90 more kangaroos in the affected area to be sure

[2/1] (d) The kangaroos have not been poisoned

## S3a (dependence of power on sample size)

A fertilizer is trialled to see if it improves crop yields. Without the fertilizer the average yield is 100, with s.d. of 10. It is expected that the fertilizer will boost yield by 3 points on average. How many plants would be needed in the treatment and control groups to be confident of demonstrating whether or not the fertilizer was effective?

[6/6] (a) 20 per group

[6/9] (b) 50 per group

[16/17] (c) 100 per group

[22/18] (d) * 300 per group

## S3b

You have been asked to test a treatment for obesity. People in the trial have a mean body mass index (BMI) of 35, with s.d. of 5. The developer argues that the treatment will reduce BMI by 2 points on average, but you are dubious as to whether it has any effect. Assuming you have a control group given a placebo and an experimental group given the treatment, what sample size should you select to give a fair test of the treatment?

[5/4] (a) x 20 per group

[16/16] (b) x 50 per group

[19/17] (c) * 100 per group

[10/13] (d) 500 per group

## S4a (likelihood of given value depends on sample size: same as S2 from parallel form, but with initial adequate N)

There are concerns that kangaroos appear unhealthy in a particular area of the Australian bush. There is concern that this may be due to eating poisoned bait that affects the blood. A naturalist has blood samples from 160 kangaroos and measures a distinctive blood marker that is lowered in poisoned animals. Previous studies have found that the mean blood marker in healthy kangaroos is 130 with s.d. of 30, whereas the mean is 120 with s.d. of 30 in poisoned animals.

The mean blood marker in the sample is 120. Assuming that there is no other explanation than poisoning for an abnormal blood marker, what can the scientist conclude?

[6/6] (a) * The kangaroos have definitely been poisoned

[12/12] (b) x The kangaroos probably have been poisoned but can't be sure. Collecting data from more kangaroos won't help

[29/30] (c) The kangaroos probably have been poisoned, but would need to collect blood from 100 more kangaroos in the affected area of bush to be sure

[3/2] (d) The kangaroos have not been poisoned

## S4b (likelihood of given value depends on sample size: same as S2 from parallel form, but with initial adequate N)

A forensic archaeologist has a set of 160 male skeletons from an ancient burial site. Experts are divided as to whether the site was colonized by Tribe A or Tribe B, the only two tribes in the region. Previous studies with large samples found that men from Tribe A had an average height of 160 cm, with s.d. of 6, and men from Tribe B had an average height of 158 cm, with s.d. of 6.

The mean height of the sample is 158 cm. What can the archaeologist conclude?

[2/0] (a) * We can be very confident that the sample comes from Tribe B

[11/6] (b) Tribe B is more likely than Tribe A, but should collect 100 more samples to be sure

[29/33] (c) Equally likely the sample comes from Tribe A or Tribe B

[8/11] (d) x Can't be sure: Tribe B is more likely than Tribe A, but collecting more samples won't help

## S5a (classic from Tversky/Kahneman law of small numbers)

In a squash tournament, the organizers are debating whether to have games of best of 3, 9 or 21 points. Holding all other rules of the game constant, if A is a slightly better player than B, which scoring will give A a better chance of winning?

[9/11] (a) best of 3 points

[6/7] (b) best of 9 points

[14/20] (c) * best of 21 points

[21/12] (d) x Won't make any difference

## S5b

Two chess players are having a tournament. They are considering whether to play the best of 5, 11 or 17 games. If player A is slightly better than player B, which tournament size should player B argue for, to get the best chance of winning?

[15/7] (a) 17 games

[8/2] (b) 11 games

[17/13] (c) * 5 games

[10/28] (d) x It doesn't matter: The chances are the same regardless of number of games

## S6a. (risk of type II error with small samples)

Two scientists are both trying to test whether a certain new drug affects hunger in mice, by giving them the drug (group D) or a placebo (group P) and then measuring their food consumption. At the start of the experiment, the average mouse eats 10 g of food pellets, with s.d. of 3 g.

Scientist A runs 10 studies with 10 mice in each group, and Scientist B runs 10 studies with 30 mice in each group. Unfortunately for them, a careless laboratory technician distributed a placebo in place of the new drug, so there should not be any effects except by chance. When the scientists look at the results, a large effect, a difference in food consumption of 3 g, is seen between the two groups (D and P) on one run of the experiment. Which scenario is more likely?

[18/20] (a) * The run with a large difference is found in the smaller group

[14/8] (b) The run with a large difference is found in the larger group

[4/4] (c) The group difference shows group D eats more than group P

[14/18] (d) x A large difference is equally likely D > P or P > D, with no effect of sample size

## S6b

Two researchers are testing the effect of oxytocin on prosocial behaviour. In their experiments, participants are given either a dose of oxytocin or a placebo. Researcher A runs 5 studies with 10 participants in each condition and Researcher B runs 5 studies with 20 participants in each condition. Unknown to the researchers, the oxytocin had been replaced by a placebo, yet a large difference (1 s.d.) is observed between two groups on one run of the experiment. Which is most likely?

[18/13] (a) * The large difference is seen in a study by the researcher using groups of 10

[7/6] (b) The large difference is seen in a study by the researcher using groups of 20

[23/23] (c) x The difference is a type II error, and could equally likely be seen with groups of 10 or 20

[2/8] (d) The difference is real—there was confusion between oxytocin and placebo.

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
