## [Peer Review File · Royal Society Open Science]

Review History

RSOS-211028.R0 (Original submission)

Review form: Reviewer 1

Do you have any ethical concerns with this paper?

No

Recommendation?

Accept with minor revision

Comments to the Author(s)

Review of: Stage 1 Registered Report: Can we shift belief in the ‘Law of Small Numbers?’
This is a registered report for an experimental manipulation designed to improve subjects’ understanding of the limitations of small samples. I believe that the research question is important and timely, as overreliance on small samples is a critical problem in biomedical research. The study also fills an important gap: Anecdotal reports suggest that teaching via simulation can help improve statistical understanding of sampling distributions (and correspondingly of “small sample neglect”), but few studies have empirically addressed this

topic. Overall, I think this is a well done study plan that provides sufficient detail for replication. The authors have also conducted two pilot studies, which helps establish study feasibility.

The authors have proposed three hypotheses: Hypothesis 1 can be viewed as a positive control (establishing the presence of “small sample neglect”). Hypothesis 2 tests whether training leads to an increase in the sample size at which subjects decide whether the effect is null or real. Hypothesis 3 tests whether there is an association between improvement on the array index (sample size when the decision is made) and improvement on the post-test quiz score. These are reasonable choices, but the authors may want to include further rationale for hypothesis 3. Why did the authors choose to focus on the association between improvement in the training task and improvement in the test score rather than simply on whether there was a post-training improvement in test score? It is possible test scores could improve following training even if training improvement and test improvement do not correlate. Maybe include H3 to test whether test scores improve and then H4 to test for associations between improvements in training and test scores?

The authors have fully described their analysis approach and statistical power calculations, and both seem reasonable. However, I would recommend that the authors provide a clear definition of the key dependent variable “array index” early in the paper. As a reader, it took multiple reads for me to parse the meaning of this variable. Under “Independent and Dependent” variables array index is defined as: “The dependent variables are mean array index per block, corresponding to the array size at which the response is made, and proportion correct for each block.” This explanation is confusing because (1) the definition of “array size” only appears later in the paper and (2) on first read, I misunderstood this sentence to say that the array index was calculated based *both* on array size and the proportion correct.

One methodological point that is unclear for me: What is the rationale for including Stopping Rules? This seems to add an unnecessary complication to the study. Since participation in the study is low-cost and low-risk, why not just collect data on the full 100?

Pilot 2 data: It would be helpful to include specific numbers from Pilot 2 to help readers gauge the validity of the sample size calculations. In Pilot 1, participants started at a mean array index value of 4.6, which leaves little room for improvement if the optimal array size is typically around 5. The sample size calculations assume a mean improvement of 1 or 2, but an improvement of 2 is impossible if participants start at 4.6; and an improvement of 1 is not expected if the goal is to optimize rather than maximize array size. What was the mean array index for block 1 for Pilot 2? Also, the Pilot 2 data don't seem to be available on OSF.

The training task deals with differences in mean values, but the some quiz questions ask about differences in proportions. Does the training actually prepare participants to answer these questions? For example, for question S2A, how would the training give participants an intuition to help them choose between $n=100$ and $n=200$ for a true difference in proportions of 17%? Also, it would be useful to have more details about how the correct and incorrect answers were selected for these questions. For example, for S2A, $N=100$ (the correct answer) corresponds to about 70% power, but for other questions the correct answer corresponds to a much higher statistical power.

Minor editorial suggestions:

-Introduction: The clarity in this paragraph could be improved: “In 1962, Cohen (1962) embarked on a project of improving psychologists’ understanding of statistical power, providing tools to help compute power and documenting the extent of underpowered studies in social psychology, with the aim of reducing waste in research efforts. However, 27 years later, Sdelmeier & Gigerenzer reported that things had not changed at all. And in 2016, similar conclusions were

drawn from a large review of studies...” It’s ambiguous as to what “things had not changed at all” and what “similar conclusions” is referring to. It would be helpful to state the problem more clearly in the first sentence, e.g.: “In 1962, Cohen documented that XX% of studies in social psychology are underpowered...”

-I think there may be an error in this description: “The prediction will be tested using linear regression: $postS \sim preS + array.diff$, where $postS$ and $preS$ are posttest and pretest scores on S-quiz items, and $array.diff$ is the difference in percent correct between the last and first block of training.” I thought that $array.diff$ was the difference in the array index between the first and last block of training, not the difference in the percent correct?

-Exploratory Analyses section: “As well as the planned analyses using percent correct as a dependent variable...” This was confusing on first read, because (1) there is more than one “percent correct” variable (e.g., percent correct in the task vs. percent correct on the quiz) and (2) only hypothesis 3 uses percent correct as the dependent variable. Maybe edit to “As well as the planned analyses described above...”

-Quiz question: “(September, November and December are all 30 days long).” In fact, December is 31 days long.

-Quiz question contains a grammatical error: “In box of 100 CDs, 40% of the disks are have been recorded by pop artists”

Review form: Reviewer 2

Do you have any ethical concerns with this paper?

No

Recommendation?

Accept with minor revision

Comments to the Author(s)

The proposed study addresses an interesting question: whether a game-like simulation can improve sample size neglect in subjects. Overall, the experiment has been carefully planned and pilot data collected to inform the design and sample size calculation.

=====

Answers to editorial questions:

1. The scientific validity of the research question(s).

> A relevant question has been postulated, motivated by, and built on, current knowledge.

2. The logic, rationale, and plausibility of the proposed hypotheses.

> The proposed hypotheses follows logically from current knowledge and is a priori highly plausible (and supported by pilot data).

3. The soundness and feasibility of the methodology and analysis pipeline (including statistical power analysis where applicable).

> Good overall, suggestions for improvement are provided below.

4. Whether the clarity and degree of methodological detail would be sufficient to replicate the proposed experimental procedures and analysis pipeline.

> Sufficient details have been provided to replicate the experiment.

5. Whether the authors provide a sufficiently clear and detailed description of the methods to prevent undisclosed flexibility in the experimental procedures or analysis pipeline.

> Sufficient details have been provided to restrict researcher degrees of freedom to reasonable levels.

6. Whether the authors have considered sufficient outcome-neutral conditions (e.g. absence of floor or ceiling effects; positive controls; other quality checks) for ensuring that the results obtained are able to test the stated hypotheses.

> Good overall, suggestions for improvement are provided below.

=====

A few points to consider:

1) Currently, there is a 2 second lag before additional data are shown and subjects must quickly decide if there is (1) evidence for an effect, (2) no evidence for an effect, or (3) insufficient data. This duration seems very short, and surely in some cases additional data are shown while subjects are trying to decide between options 1 and 2. In other words, a decision is made for them if they are not fast enough. This also differs from the real world where researchers can stare at their data as long as they like and then actively decide to collect more. Would it be better to allow subjects to click a button to see more data (or decide between options 1 and 2 given the current data)? This would remove speed of processing/decision making from the picture.

2) Analysis Plan: "Subjects who score 90% or more correct in block 1 will be removed from the main analysis and treated separately." I would strongly recommend against removing these subjects since regression to the mean becomes a problem. Subjects that score high in Block 1 are those that have a good prior understanding of sample size effects but also those with a "positive residual" -- in other words their observed score is higher than their true understanding because they got lucky. Assume that the training has no effect. When removing subjects with >90% in Block 1, the Block 4 scores will be higher than Block 1 scores. This occurs because high values are removed from Block 1 thus lowering the mean. And, those with negative residuals in Block 1 regress to the mean and improve, while those with positive residuals in Block 1 cannot regress to lower values since they have been removed. Thus the estimated treatment effect will be a mix of the true treatment effect plus regression to the mean -- both pointing in the same direction. Hence, it's better to retain all subjects and have an uncontaminated treatment effect (the sample size can be increased if there is a loss of power).

If weeding out those who have a good initial understanding of sample size effects is essential, it would be better to use the pretest quiz as a filter, since positive residuals in the quiz will be uncorrelated with positive residuals in Block 1 (but hopefully performance on the quiz correlates with performance in Block 1).

3) Currently, only data from Blocks 1 and 4 are used for analysis, which ignores 50% of the collected data (Blocks 2 and 3). A simple alternative analysis would be to regress the outcome on block for each subject and calculate the slope of the regression line (outcome for each block vs block number). The collection of slopes could then be tested if they differ from zero (or substituted for "array.diff" in other models). This may not be appropriate if the learning rate across the four blocks is not approximately linear. Such an approach may provide more stable results since information from two additional blocks are used. The pilot data can indicate if this method has merit.

4) Would the dollar amount won be a useful outcome variable as it integrates both the correctness and array index information?

Typos:

Page 11, line 28: Should be "lose 4 points"?

Page 11, line 32: Should be "likelihood of one scenario"?

Decision letter (RSOS-211028.R0)

Dear Professor Bishop,

On behalf of the Editors, I am pleased to inform you that your Manuscript RSOS-211028 entitled "Stage 1 Registered Report: Can we shift belief in the 'Law of Small Numbers?'" deemed suitable for in-principle acceptance in Royal Society Open Science subject to minor revision in accordance with the referee and editor suggestions. Please find their comments at the end of this email.

The reviewers and handling editors have recommended publication, but also suggest some minor revisions to your manuscript. Therefore, I invite you to respond to the comments and revise your manuscript.

Please you submit the revised version of your manuscript within 14 days. If you do not think you will be able to meet this date please let us know immediately.

Full author guidelines can be found here <https://royalsocietypublishing.org/rsos/registered-reports#ReviewerGuideRegRep>.

Once again, thank you for submitting your manuscript to Royal Society Open Science and we look forward to receiving your revision. If you have any questions at all, please do not hesitate to get in touch.

Best regards,
Lianne Parkhouse
Editorial Coordinator

on behalf of Professor Chris Chambers
(Subject Editor, Royal Society Open Science)
openscience@royalsociety.org

Associate Editor Comments to Author (Professor Chris Chambers):

Two expert reviewers have now assessed the Stage 1 manuscript. As you will see, both are constructive and enthusiastic, and recommend IPA following moderate revisions. Reviewer 1 asks mainly for changes to improve clarity and provided missing study details/context, while Reviewer 2 raises a number of methodological concerns that will need to be addressed -- including, most importantly, a concern regarding regression to the mean. Please respond thoroughly to all points raised.

Reviewer comments to Author:

Reviewer: 1

Comments to the Author(s)

Review of: Stage 1 Registered Report: Can we shift belief in the 'Law of Small Numbers?' This is a registered report for an experimental manipulation designed to improve subjects' understanding of the limitations of small samples. I believe that the research question is important and timely, as overreliance on small samples is a critical problem in biomedical research. The study also fills an important gap: Anecdotal reports suggest that teaching via simulation can help improve statistical understanding of sampling distributions (and correspondingly of "small sample neglect"), but few studies have empirically addressed this topic. Overall, I think this is a well done study plan that provides sufficient detail for replication. The authors have also conducted two pilot studies, which helps establish study feasibility.

The authors have proposed three hypotheses: Hypothesis 1 can be viewed as a positive control (establishing the presence of "small sample neglect"). Hypothesis 2 tests whether training leads to an increase in the sample size at which subjects decide whether the effect is null or real. Hypothesis 3 tests whether there is an association between improvement on the array index (sample size when the decision is made) and improvement on the post-test quiz score. These are reasonable choices, but the authors may want to include further rationale for hypothesis 3. Why did the authors choose to focus on the association between improvement in the training task and improvement in the test score rather than simply on whether there was a post-training improvement in test score? It is possible test scores could improve following training even if training improvement and test improvement do not correlate. Maybe include H3 to test whether test scores improve and then H4 to test for associations between improvements in training and test scores?

The authors have fully described their analysis approach and statistical power calculations, and both seem reasonable. However, I would recommend that the authors provide a clear definition of the key dependent variable "array index" early in the paper. As a reader, it took multiple reads for me to parse the meaning of this variable. Under "Independent and Dependent" variables array index is defined as: "The dependent variables are mean array index per block, corresponding to the array size at which the response is made, and proportion correct for each block." This explanation is confusing because (1) the definition of "array size" only appears later in the paper and (2) on first read, I misunderstood this sentence to say that the array index was calculated based *both* on array size and the proportion correct.

One methodological point that is unclear for me: What is the rationale for including Stopping Rules? This seems to add an unnecessary complication to the study. Since participation in the study is low-cost and low-risk, why not just collect data on the full 100?

Pilot 2 data: It would be helpful to include specific numbers from Pilot 2 to help readers gauge the validity of the sample size calculations. In Pilot 1, participants started at a mean array index value of 4.6, which leaves little room for improvement if the optimal array size is typically around 5. The sample size calculations assume a mean improvement of 1 or 2, but an improvement of 2 is impossible if participants start at 4.6; and an improvement of 1 is not expected if the goal is to optimize rather than maximize array size. What was the mean array index for block 1 for Pilot 2? Also, the Pilot 2 data don't seem to be available on OSF.

The training task deals with differences in mean values, but the some quiz questions ask about differences in proportions. Does the training actually prepare participants to answer these questions? For example, for question S2A, how would the training give participants an intuition to help them choose between $n=100$ and $n=200$ for a true difference in proportions of 17%? Also, it would be useful to have more details about how the correct and incorrect answers were selected for these questions. For example, for S2A, $N=100$ (the correct answer) corresponds to about 70% power, but for other questions the correct answer corresponds to a much higher statistical power.

Minor editorial suggestions:

-Introduction: The clarity in this paragraph could be improved: "In 1962, Cohen (1962) embarked on a project of improving psychologists' understanding of statistical power, providing tools to help compute power and documenting the extent of underpowered studies in social psychology, with the aim of reducing waste in research efforts. However, 27 years later, Sdelmeier & Gigerenzer reported that things had not changed at all. And in 2016, similar conclusions were drawn from a large review of studies..." It's ambiguous as to what "things had not changed at all" and what "similar conclusions" is referring to. It would be helpful to state the problem more clearly in the first sentence, e.g.: "In 1962, Cohen documented that XX% of studies in social psychology are underpowered..."

-I think there may be an error in this description: "The prediction will be tested using linear regression: $\text{postS} \sim \text{preS} + \text{array.diff}$, where postS and preS are posttest and pretest scores on S-quiz items, and array.diff is the difference in percent correct between the last and first block of training." I thought that array.diff was the difference in the array index between the first and last block of training, not the difference in the percent correct?

-Exploratory Analyses section: "As well as the planned analyses using percent correct as a dependent variable..." This was confusing on first read, because (1) there is more than one "percent correct" variable (e.g., percent correct in the task vs. percent correct on the quiz) and (2) only hypothesis 3 uses percent correct as the dependent variable. Maybe edit to "As well as the planned analyses described above..."

-Quiz question: "(September, November and December are all 30 days long)." In fact, December is 31 days long.

-Quiz question contains a grammatical error: "In box of 100 CDs, 40% of the disks are have been recorded by pop artists"

Reviewer: 2

Comments to the Author(s)

The proposed study addresses an interesting question: whether a game-like simulation can improve sample size neglect in subjects. Overall, the experiment has been carefully planned and pilot data collected to inform the design and sample size calculation.

Answers to editorial questions:

1. The scientific validity of the research question(s).
 - > A relevant question has been postulated, motivated by, and built on, current knowledge.
2. The logic, rationale, and plausibility of the proposed hypotheses.
 - > The proposed hypotheses follows logically from current knowledge and is a priori highly plausible (and supported by pilot data).
3. The soundness and feasibility of the methodology and analysis pipeline (including statistical power analysis where applicable).
 - > Good overall, suggestions for improvement are provided below.
4. Whether the clarity and degree of methodological detail would be sufficient to replicate the proposed experimental procedures and analysis pipeline.
 - > Sufficient details have been provided to replicate the experiment.
5. Whether the authors provide a sufficiently clear and detailed description of the methods to prevent undisclosed flexibility in the experimental procedures or analysis pipeline.
 - > Sufficient details have been provided to restrict researcher degrees of freedom to reasonable levels.
6. Whether the authors have considered sufficient outcome-neutral conditions (e.g. absence of floor or ceiling effects; positive controls; other quality checks) for ensuring that the results obtained are able to test the stated hypotheses.
 - > Good overall, suggestions for improvement are provided below.

A few points to consider:

1) Currently, there is a 2 second lag before additional data are shown and subjects must quickly decide if there is (1) evidence for an effect, (2) no evidence for an effect, or (3) insufficient data. This duration seems very short, and surely in some cases additional data are shown while subjects are trying to decide between options 1 and 2. In other words, a decision is made for them if they are not fast enough. This also differs from the real world where researchers can stare at their data as long as they like and then actively decide to collect more. Would it be better to allow subjects to click a button to see more data (or decide between options 1 and 2 given the current data)? This would remove speed of processing/decision making from the picture.

2) Analysis Plan: "Subjects who score 90% or more correct in block 1 will be removed from the main analysis and treated separately." I would strongly recommend against removing these subjects since regression to the mean becomes a problem. Subjects that score high in Block 1 are those that have a good prior understanding of sample size effects but also those with a "positive residual" -- in other words their observed score is higher than their true understanding because they got lucky. Assume that the training has no effect. When removing subjects with >90% in Block 1, the Block 4 scores will be higher than Block 1 scores. This occurs because high values are removed from Block 1 thus lowering the mean. And, those with negative residuals in Block 1 regress to the mean and improve, while those with positive residuals in Block 1 cannot regress to lower values since they have been removed. Thus the estimated treatment effect will be a mix of the true treatment effect plus regression to the mean -- both pointing in the same direction.

Hence, it's better to retain all subjects and have an uncontaminated treatment effect (the sample size can be increased if there is a loss of power).

If weeding out those who have a good initial understanding of sample size effects is essential, it would be better to use the pretest quiz as a filter, since positive residuals in the quiz will be uncorrelated with positive residuals in Block 1 (but hopefully performance on the quiz correlates with performance in Block 1).

3) Currently, only data from Blocks 1 and 4 are used for analysis, which ignores 50% of the collected data (Blocks 2 and 3). A simple alternative analysis would be to regress the outcome on block for each subject and calculate the slope of the regression line (outcome for each block vs block number). The collection of slopes could then be tested if they differ from zero (or substituted for "array.diff" in other models). This may not be appropriate if the learning rate across the four blocks is not approximately linear. Such an approach may provide more stable results since information from two additional blocks are used. The pilot data can indicate if this method has merit.

4) Would the dollar amount won be a useful outcome variable as it integrates both the correctness and array index information?

Typos:

Page 11, line 28: Should be "lose 4 points"?

Page 11, line 32: Should be "likelihood of one scenario"?

Author's Response to Decision Letter for (RSOS-211028.R0)

See Appendix A.

RSOS-211028.R1 (Revision)

Review form: Reviewer 1

Do you have any ethical concerns with this paper?

No

Recommendation?

Accept in principle

Comments to the Author(s)

The authors have done an excellent job on the revision. Earning score is a better choice for the dependent variable, and I appreciate the improved clarity in the protocol. I have no further concerns.

Review form: Reviewer 2

Do you have any ethical concerns with this paper?

No

Recommendation?

Accept in principle

Comments to the Author(s)

It is indeed interesting that people do not learn when they have more time to decide, and the likely explanation is that training effectiveness depends on exposure to increasing arrays -- as the authors suggest. I therefore think their suggested/current approach is preferred.

Decision letter (RSOS-211028.R1)

Dear Professor Bishop

On behalf of the Editor, I am pleased to inform you that your Manuscript RSOS-211028.R1 entitled "Stage 1 Registered Report: Can we shift belief in the 'Law of Small Numbers?'" has been accepted in principle for publication in Royal Society Open Science. The reviewers' and editors' comments are included at the end of this email.

You may now progress to Stage 2 and complete the study as approved. Before commencing data collection we ask that you:

- 1) Update the journal office as to the anticipated completion date of your study.
- 2) Register your approved protocol on the Open Science Framework (<https://osf.io/>) or other recognised repository, either publicly or privately under embargo until submission of the Stage 2 manuscript. Please note that a time-stamped, independent registration of the protocol is mandatory under journal policy, and manuscripts that do not conform to this requirement cannot be considered at Stage 2. The protocol should be registered unchanged from its current approved state, with the time-stamp preceding implementation of the approved study design.

Following completion of your study, we invite you to resubmit your paper for peer review as a Stage 2 Registered Report. Please note that your manuscript can still be rejected for publication at Stage 2 if the Editors consider any of the following conditions to be met:

- The results were unable to test the authors' proposed hypotheses by failing to meet the approved outcome-neutral criteria.
- The authors altered the Introduction, rationale, or hypotheses, as approved in the Stage 1 submission.
- The authors failed to adhere closely to the registered experimental procedures. Please note that any deviations from the approved experimental procedures must be communicated to the editor immediately for approval, and prior to the completion of data collection. Failure to do so can

result in revocation of in-principle acceptance and rejection at Stage 2 (see complete guidelines for further information).

- Any post-hoc (unregistered) analyses were either unjustified, insufficiently caveated, or overly dominant in shaping the authors' conclusions.
- The authors' conclusions were not justified given the data obtained.

We encourage you to read the complete guidelines for authors concerning Stage 2 submissions at <https://royalsocietypublishing.org/rsos/registered-reports#ReviewerGuideRegRep>. Please especially note the requirements for data sharing, reporting the URL of the independently registered protocol, and that withdrawing your manuscript will result in publication of a Withdrawn Registration.

Once again, thank you for submitting your manuscript to Royal Society Open Science and we look forward to receiving your Stage 2 submission. If you have any questions at all, please do not hesitate to get in touch. We look forward to hearing from you shortly with the anticipated submission date for your stage two manuscript.

on behalf of Professor Chris Chambers (Registered Reports Editor, Royal Society Open Science)
openscience@royalsociety.org

Associate Editor Comments to Author (Professor Chris Chambers):

The reviewers are now satisfied, and based on their evaluations and my own reading, the manuscript is now suitable for IPA.

Reviewers' comments to Author:

Reviewer: 1

Comments to the Author(s)

The authors have done an excellent job on the revision. Earning score is a better choice for the dependent variable, and I appreciate the improved clarity in the protocol. I have no further concerns.

Reviewer: 2

Comments to the Author(s)

It is indeed interesting that people do not learn when they have more time to decide, and the likely explanation is that training effectiveness depends on exposure to increasing arrays -- as the authors suggest. I therefore think their suggested/current approach is preferred.

Author's Response to Decision Letter for (RSOS-211028.R1)

See Appendix B.

RSOS-211028.R2

Review form: Reviewer 1

Is the manuscript scientifically sound in its present form?

Yes

Are the interpretations and conclusions justified by the results?

Yes

Is the language acceptable?

Yes

Do you have any ethical concerns with this paper?

No

Recommendation?

Major revision

Comments to the Author(s)

The data collected are able to test the authors' hypotheses.

The introduction, rationale and stated hypotheses are the same as the approved Stage 1 submission.

The authors adhered to the registered experimental procedures. As they disclosed, they failed to randomly intersperse the S and P-items on the quiz for the first 50 participants. But they fixed this issue for the second set of 50 participants, and the oversight did not appear to affect the results.

The exploratory analysis of the individual S-items on the quiz was helpful and warranted. Given the content of the quiz questions, I'm not surprised that the beeswarm task failed to improve performance on the quiz. Some of the questions require quite a bit of extrapolation from the beeswarm task.

I didn't find Table 8 particularly useful – comparing the "aware" and "unaware" participants on baseline characteristics was not terribly informative, and I would recommend dropping this.

I believe that the authors' main conclusions are warranted given their data. The data demonstrate some improvement in the beeswarm task itself but little improvement in quiz performance.

I do have some comments to improve clarity:

1. I found the abstract too vague. For example, "there were significant gains on the training."

What does this mean? What improved and by how much? Similarly, "there was no generalisation to understanding of quiz items." What does this mean specifically? That participants did not have an increase in their quiz score? Readers need some of these details up front in the abstract.

2. Tables and figures should stand alone and not require the reader to consult the text. However, almost all the tables and figures contained insufficient information. For example:

-Table 4: Statistics self-rating is on what scale? Understanding T and understanding power are measured on what scale?

-Table 5: What is "item type"? Is this P-item versus S-item or S-item versus P-item? What is pre-post? Is this pre versus post or post versus pre? Needs to be clear without having to read the text.

-Table 6: Is very hard to understand without consulting the text. What is "bias"? What is "other1"? What is "other2"? Were these errors from Sample 1 or Sample 2 or both?

-Figure 8: How many are in the aware and unaware subgroups? What are the sample sizes? Also, what are the "mean array index", "d prime", "absolute log likelihood at point of response", and "absolute observed effect size at point of response"? This figure again requires the reader to consult the text to understand these outcomes. Some explanation should be given within the figure itself.

3. Some text is also hard to follow:

-When describing the pre-training and post-training means on the S-items quiz, I found the descriptions confusing. The authors switch between percentages (e.g., 83%) and decimals (e.g. 0.28), which is confusing. Also, why not present the raw numbers correct rather than the percentages? As a reader, I would find it much clearer and more transparent to learn that the average score was 1.68 out of 6 rather than 28%.

-I also found this description confusing: "The proportion of participants scoring less than 50% on the pre-training P-items was 0.34 for Sample 1 and 0.36 for Sample 2." Do you mean that 34% scored less than 50%? This is confusing. Why not say, e.g, 34% got less than 3 out of 6 correct on the P-items? Again, focusing on the raw number correct seems much clearer than focusing on the percentage correct.

-"We divided the self-reported responses into those that did and did not indicate awareness that larger sample sizes (or longer waiting) gives more reliable estimates." How many were in each group? This should be made explicit on first mention.

4. I'd recommend adding a graph (histogram) that shows the distribution of correct answers on the S-items (0-6) before and after the training; or the distribution of the change in number correct (e.g., -1, +1, 0, etc.). This would be a very helpful visualization for the reader to see that quiz scores did not improve.

Review form: Reviewer 2

Is the manuscript scientifically sound in its present form?

Yes

Are the interpretations and conclusions justified by the results?

Yes

Is the language acceptable?

Yes

Do you have any ethical concerns with this paper?

No

Recommendation?

Accept as is

Comments to the Author(s)

Answers to editorial questions:

1) Whether the data are able to test the authors' proposed hypotheses by passing the approved outcome-neutral criteria (such as absence of floor and ceiling effects or success of positive controls)

> Yes, the data can test the hypotheses/predictions. Data from the 3 pilot studies provided enough information to know what the data from the final experiment would look like, and there were no surprises.

2) Whether the Introduction, rationale and stated hypotheses are the same as the approved Stage 1 submission

> The hypotheses/predictions remain the same as Stage 1.

3) Whether the authors adhered precisely to the registered experimental procedures

> The authors adhered to the registered procedures. There was the issue of 50 subjects (Sample 1) that had quiz items blocked rather than randomised, which deviated from the protocol, but this should not affect the results and conclusions. Indeed, the Sample 1 and Sample 2 results were analysed and presented separately and they were similar.

4) Where applicable, whether any unregistered exploratory statistical analyses are justified, methodologically sound, and informative

> N/A

5) Whether the authors' conclusions are justified given the data

> Yes, the conclusions are justified by the data.

I have no further comments on the manuscript, other than one typo:

Page 33, Line 6: should be "...*if* A is a better player..."

Decision letter (RSOS-211028.R2)

Dear Professor Bishop:

On behalf of the Editor, I am pleased to inform you that your Stage 2 Registered Report RSOS-211028.R2 entitled "Stage 2 Registered Report: Can we shift belief in the 'Law of Small Numbers'?" has been deemed suitable for publication in Royal Society Open Science subject to minor revision in accordance with the referee suggestions. Please find the referees' comments at the end of this email.

The reviewers and Subject Editor have recommended publication, but also suggest some minor revisions to your manuscript. We invite you to respond to the comments and revise your manuscript. Below the referees' and Editors' comments (where applicable) we provide additional requirements. Final acceptance of your manuscript is dependent on these requirements being met. We provide guidance below to help you prepare your revision.

Please submit your revised manuscript and required files (see below) no later than 7 days from today's (ie 07-Jan-2022) date. Note: the ScholarOne system will 'lock' if submission of the revision is attempted 7 or more days after the deadline. If you do not think you will be able to meet this deadline please contact the editorial office immediately.

Please note article processing charges apply to papers accepted for publication in Royal Society Open Science (<https://royalsocietypublishing.org/rsos/charges>). Charges will also apply to papers transferred to the journal from other Royal Society Publishing journals, as well as papers submitted as part of our collaboration with the Royal Society of Chemistry

(<https://royalsocietypublishing.org/rsos/chemistry>). Fee waivers are available but must be requested when you submit your revision (<https://royalsocietypublishing.org/rsos/waivers>).

on behalf of Professor Chris Chambers (Associate Editor) and Chris Chambers
 (Registered Reports Editor, Royal Society Open Science)
openscience@royalsociety.org

Associate Editor Comments to Author (Professor Chris Chambers):

Associate Editor: 1

Comments to the Author:

The two reviewers who assessed the Stage 1 submission kindly returned to evaluate the completed manuscript with results. As you will see, the comments are overall very encouraging. Reviewer 1 offers some useful suggestions for improving clarity of the presentation, while Reviewer 2 is satisfied with the manuscript as-is. In revising, please minimise all changes to the accepted Stage 1 parts of the manuscript, unless amendments are necessary to avoid confusion or correct factual errors (this constraint does not apply to the Abstract). Provided you are able to address the comments in a minor revision, final Stage 1 acceptance should be forthcoming without further in-depth review.

Comments to Author:

Reviewer: 1

Comments to the Author(s)

The data collected are able to test the authors' hypotheses.

The introduction, rationale and stated hypotheses are the same as the approved Stage 1 submission.

The authors adhered to the registered experimental procedures. As they disclosed, they failed to randomly intersperse the S and P-items on the quiz for the first 50 participants. But they fixed this issue for the second set of 50 participants, and the oversight did not appear to affect the results. The exploratory analysis of the individual S-items on the quiz was helpful and warranted. Given the content of the quiz questions, I'm not surprised that the beeswarm task failed to improve performance on the quiz. Some of the questions require quite a bit of extrapolation from the beeswarm task.

I didn't find Table 8 particularly useful – comparing the “aware” and “unaware” participants on baseline characteristics was not terribly informative, and I would recommend dropping this. I believe that the authors' main conclusions are warranted given their data. The data demonstrate some improvement in the beeswarm task itself but little improvement in quiz performance.

I do have some comments to improve clarity:

1. I found the abstract too vague. For example, “there were significant gains on the training.” What does this mean? What improved and by how much? Similarly, “there was no generalisation to understanding of quiz items.” What does this mean specifically? That participants did not have an increase in their quiz score? Readers need some of these details up front in the abstract.

2. Tables and figures should stand alone and not require the reader to consult the text. However, almost all the tables and figures contained insufficient information. For example:

-Table 4: Statistics self-rating is on what scale? Understanding T and understanding power are measured on what scale?

-Table 5: What is "item type"? Is this P-item versus S-item or S-item versus P-item? What is pre-post? Is this pre versus post or post versus pre? Needs to be clear without having to read the text.

-Table 6: Is very hard to understand without consulting the text. What is "bias"? What is "other1"? What is "other2"? Were these errors from Sample 1 or Sample 2 or both?

-Figure 8: How many are in the aware and unaware subgroups? What are the sample sizes? Also, what are the "mean array index", "d prime", "absolute log likelihood at point of response", and "absolute observed effect size at point of response"? This figure again requires the reader to consult the text to understand these outcomes. Some explanation should be given within the figure itself.

3. Some text is also hard to follow:

-When describing the pre-training and post-training means on the S-items quiz, I found the descriptions confusing. The authors switch between percentages (e.g., 83%) and decimals (e.g. 0.28), which is confusing. Also, why not present the raw numbers correct rather than the percentages? As a reader, I would find it much clearer and more transparent to learn that the average score was 1.68 out of 6 rather than 28%.

-I also found this description confusing: "The proportion of participants scoring less than 50% on the pre-training P-items was 0.34 for Sample 1 and 0.36 for Sample 2." Do you mean that 34% scored less than 50%? This is confusing. Why not say, e.g, 34% got less than 3 out of 6 correct on the P-items? Again, focusing on the raw number correct seems much clearer than focusing on the percentage correct.

-"We divided the self-reported responses into those that did and did not indicate awareness that larger sample sizes (or longer waiting) gives more reliable estimates." How many were in each group? This should be made explicit on first mention.

4. I'd recommend adding a graph (histogram) that shows the distribution of correct answers on the S-items (0-6) before and after the training; or the distribution of the change in number correct (e.g., -1, +1, 0, etc.). This would be a very helpful visualization for the reader to see that quiz scores did not improve.

Reviewer: 2

Comments to the Author(s)

Answers to editorial questions:

1) Whether the data are able to test the authors' proposed hypotheses by passing the approved outcome-neutral criteria (such as absence of floor and ceiling effects or success of positive controls)

> Yes, the data can test the hypotheses/predictions. Data from the 3 pilot studies provided enough information to know what the data from the final experiment would look like, and there were no surprises.

2) Whether the Introduction, rationale and stated hypotheses are the same as the approved Stage 1 submission

> The hypotheses/predictions remain the same as Stage 1.

3) Whether the authors adhered precisely to the registered experimental procedures

> The authors adhered to the registered procedures. There was the issue of 50 subjects (Sample 1) that had quiz items blocked rather than randomised, which deviated from the protocol, but this should not affect the results and conclusions. Indeed, the Sample 1 and Sample 2 results were analysed and presented separately and they were similar.

4) Where applicable, whether any unregistered exploratory statistical analyses are justified, methodologically sound, and informative

> N/A

5) Whether the authors' conclusions are justified given the data

> Yes, the conclusions are justified by the data.

I have no further comments on the manuscript, other than one typo:

Page 33, Line 6: should be "...*if* A is a better player..."

===PREPARING YOUR MANUSCRIPT===

one version should clearly identify all the changes that have been made (for instance, in coloured highlight, in bold text, or tracked changes);

===PREPARING YOUR REVISION IN SCHOLARONE===

-- If you are requesting an article processing charge waiver, you must select the relevant waiver option (if requesting a discretionary waiver, the form should have been uploaded, see 'File upload' above).

-- If you have uploaded any electronic supplementary (ESM) files, please ensure you follow the guidance at <https://royalsociety.org/journals/authors/author-guidelines/#supplementary-material> to include a suitable title and informative caption. An example of appropriate titling and captioning may be found at https://figshare.com/articles/Table_S2_from_Is_there_a_trade-off_between_peak_performance_and_performance_breadth_across_temperatures_for_aerobic_scope_in_teleost_fishes_/3843624.

Author's Response to Decision Letter for (RSOS-211028.R2)

See Appendix C.

Decision letter (RSOS-211028.R3)

Dear Professor Bishop:

It is a pleasure to accept your Stage 2 Registered Report entitled "Can we shift belief in the 'Law of Small Numbers'?" in its current form for publication in Royal Society Open Science.

Thank you for your fine contribution. On behalf of the Editors of Royal Society Open Science, we look forward to your continued contributions to the journal.

Kind regards,
Royal Society Open Science Editorial Office

on behalf of Professor Chris Chambers (Subject Editor)
openscience@royalsociety.org

Appendix A

We thank the reviewers for their thoughtful responses. Although we were told that only minor revisions were required, we felt that some of the points raised required more major changes, and hence we requested an extension to your deadline. The reviewer comments were enormously helpful in provoking discussions about the optimal approach, and made us rethink some aspects of the analysis.

We will document specific responses to reviewers below, but the main things we have done are to:

- a) Devise and gather pilot data on a 3rd version of the task, based on the suggestion by reviewer 2. This was very useful, but results indicated that we would be unwise to modify the task in this way, as we did not observe any learning with this version.
- b) Create a new R markdown document that organises and summarises the data from all three pilot studies, which is available on OSF (<https://osf.io/s39qd/>), together with the pilot data.
- c) Revise the simulation for the power analysis to make it compatible with the current version of the analysis.
- d) Respecify hypotheses, taking into account reviewer feedback.

Reviewer: 1

1.1 The authors have proposed three hypotheses: Hypothesis 1 can be viewed as a positive control (establishing the presence of “small sample neglect”). Hypothesis 2 tests whether training leads to an increase in the sample size at which subjects decide whether the effect is null or real. Hypothesis 3 tests whether there is an association between improvement on the array index (sample size when the decision is made) and improvement on the post-test quiz score. These are reasonable choices, but the authors may want to include further rationale for hypothesis 3. Why did the authors choose to focus on the association between improvement in the training task and improvement in the test score rather than simply on whether there was a post-training improvement in test score? It is possible test scores could improve following training even if training improvement and test improvement do not correlate. Maybe include H3 to test whether test scores improve and then H4 to test for associations between improvements in training and test scores?

Response: The linear regression analysis we propose does allow us to test the significance of an overall pre/post timing effect as well as an interaction with training gain, but our primary prediction is that we will see improvement only in those who show benefit from training - hence the focus on the interaction for testing H3.

We had, in table 2, shown power to detect an overall gain in quiz score, and noted it is relatively poor except when we assume a 2 point quiz gain and have a large sample, with at least 33% of subjects showing learning. Because we predict only a subset of subjects (usually a minority) will learn, any overall impact of pre vs post testing will be diluted by non-learners. Indeed, if we see an improvement in those who don't show a training gain, this would not give good evidence for training, but would rather suggest any improvement just reflected the impact of familiarity with the quiz format. We have now made it clearer that

both the overall effect of pre-post testing, as well as the interaction with learner status, are obtained from the regression analysis.

Note also that, in summarising the pilot data, we realised that we could do more with the self-report data from participants about the strategies they used. Accordingly, we have added a further prediction that those who report that they became aware of the importance of sample size during training will show better learning than those who do not. Tentative evidence for this was seen in Pilot 2.

1.2 The authors have fully described their analysis approach and statistical power calculations, and both seem reasonable. However, I would recommend that the authors provide a clear definition of the key dependent variable “array index” early in the paper. As a reader, it took multiple reads for me to parse the meaning of this variable. Under “Independent and Dependent” variables array index is defined as: “The dependent variables are mean array index per block, corresponding to the array size at which the response is made, and proportion correct for each block.” This explanation is confusing because (1) the definition of “array size” only appears later in the paper and (2) on first read, I misunderstood this sentence to say that the array index was calculated based *both* on array size and the proportion correct.

Response: We have revised to clarify this, also taking into account the decision to use mean earnings per block as a dependent variable (see reviewer 2, point 2.4).

1.3 One methodological point that is unclear for me: What is the rationale for including Stopping Rules? This seems to add an unnecessary complication to the study. Since participation in the study is low-cost and low-risk, why not just collect data on the full 100?

Response: We have not adopted this suggestion, as we prefer to use stopping rules on grounds of efficiency. Although the project is relatively low-cost, we do pay participants around £10 each for participation, and so if we can get a clear result with 50 rather than 100 participants, we save £500. It is trivially easy to check results at each stopping point, given that we have analysis scripts prepared.

1.4 Pilot 2 data: It would be helpful to include specific numbers from Pilot 2 to help readers gauge the validity of the sample size calculations. In Pilot 1, participants started at a mean array index value of 4.6, which leaves little room for improvement if the optimal array size is typically around 5. The sample size calculations assume a mean improvement of 1 or 2, but an improvement of 2 is impossible if participants start at 4.6; and an improvement of 1 is not expected if the goal is to optimize rather than maximize array size. What was the mean array index for block 1 for Pilot 2? Also, the Pilot 2 data don't seem to be available on OSF.

Response: Apologies - we had deposited the raw data and the script to process it, but not the summary data. We have now added a further pilot study and provided a detailed report on all three pilot studies.

In scrutinising the data from pilot 2, however, we found that array index means were quite high, even for block 1. The mean value was significantly smaller than 5, but nevertheless above 4. We had hoped that we might see a clearer effect of increasing array index over

blocks with a larger sample, especially given the encouraging evidence of learning from the percentage correct values, and the self-report evidence that subjects were becoming aware of array size issues. But the finding that many subjects don't start with small array sizes is contrary to expectation, and this also limits the amount of increase in array size that could be seen.

We anticipated that modifying the task to allow more time to respond (see point 2.1 below, Pilot 3) might shift the mean array index down in early blocks, but although this was to some extent effective, there was barely any increase in array index from initial to final blocks. On the basis of further simulations, we realised that array index is simply too blunt a measure to detect learning: even when participants reported starting to use it, the mean array index was not obviously different in later blocks. We discuss further below our decision to follow the suggestion of reviewer 2 that we move to a measure of learning that reflects accuracy.

1.5 The training task deals with differences in mean values, but some quiz questions ask about differences in proportions. Does the training actually prepare participants to answer these questions? For example, for question S2A, how would the training give participants an intuition to help them choose between $n=100$ and $n=200$ for a true difference in proportions of 17%? Also, it would be useful to have more details about how the correct and incorrect answers were selected for these questions. For example, for S2A, $N=100$ (the correct answer) corresponds to about 70% power, but for other questions the correct answer corresponds to a much higher statistical power.

Response: This is a good point - where we are talking about proportions above a cutoff, then this really translates to variances, because the bigger the variance, the more fall above the cutoff. So we hoped we were using questions that addressed whether the person had got a conceptual understanding of the impact of sample size on variability of estimates. But we agree that may be too subtle, and, as we do not want to reduce the chances of showing a training effect, it would be better to have questions more tightly linked to the training. We have therefore substituted questions that are more focused on sample size influence on means comparisons for quiz items S2 and S4.

Just as an aside, though, we do include relevant calculations of answers for quiz items in the Rmd script for this manuscript, and these gave power over 90% for the question that was queriedL

```
# Effect size for difference in proportions is 2*arcsin(sqrt(p1))-2*arcsin(sqrt(p2))
```

```
h <- 2*asin(sqrt(p1))-2*asin(sqrt(p2))  
pwr.p.test(h=h,n=100,sig.level=.05)
```

proportion power calculation for binomial distribution (arcsine transformation)

```
h = 0.3469169  
n = 100  
sig.level = 0.05  
power = 0.9343768  
alternative = two.sided
```

Minor editorial suggestions:

1.6 -Introduction: The clarity in this paragraph could be improved: "In 1962, Cohen (1962) embarked on a project of improving psychologists' understanding of statistical power, providing tools to help compute power and documenting the extent of underpowered studies in social psychology, with the aim of reducing waste in research efforts. However, 27 years later, Sdelmeier & Gigerenzer reported that things had not changed at all. And in 2016, similar conclusions were drawn from a large review of studies..." It's ambiguous as to what "things had not changed at all" and what "similar conclusions" is referring to. It would be helpful to state the problem more clearly in the first sentence, e.g.: "In 1962, Cohen documented that XX% of studies in social psychology are underpowered..."

Response: Change has been made as suggested. The passage now reads:

In 1962, Cohen (1962) embarked on a project of improving psychologists' understanding of statistical power, providing tools to help people compute power and documenting the extent of underpowered studies in social psychology, with the aim of reducing waste in research efforts. He analysed 70 studies published in the Journal of Abnormal and Social Psychology and found that mean power to detect small effects was 0.18, to detect medium effects was 0.48 and to detect large effects was 0.83. Given that most effects in this field are small or medium, this indicated serious limitations of study design.

1.7 -I think there may be an error in this description: "The prediction will be tested using linear regression: $\text{postS} \sim \text{preS} + \text{array.diff}$, where postS and preS are posttest and pretest scores on S-quiz items, and array.diff is the difference in percent correct between the last and first block of training." I thought that array.diff was the difference in the array index between the first and last block of training, not the difference in the percent correct?

Response: Thanks for pointing this out. Indeed, it reflects a point of uncertainty about the optimal analysis. We have now settled on using a measure reflecting accuracy as the dependent variable, and amended the text accordingly.

1.8-Exploratory Analyses section: "As well as the planned analyses using percent correct as a dependent variable..." This was confusing on first read, because (1) there is more than one "percent correct" variable (e.g., percent correct in the task vs. percent correct on the quiz) and (2) only hypothesis 3 uses percent correct as the dependent variable. Maybe edit to "As well as the planned analyses described above..."

Response: Agree this was very confusing. Now revised and hopefully clearer.

1.9 -Quiz question: "(September, November and December are all 30 days long)." In fact, December is 31 days long.

Response: Thanks for flagging this up. Now corrected by using a different month.

1.10 -Quiz question contains a grammatical error: "In box of 100 CDs, 40% of the disks are have been recorded by pop artists"

Response. Corrected

Reviewer: 2

2.1 Currently, there is a 2 second lag before additional data are shown and subjects must quickly decide if there is (1) evidence for an effect, (2) no evidence for an effect, or (3) insufficient data. This duration seems very short, and surely in some cases additional data are shown while subjects are trying to decide between options 1 and 2. In other words, a decision is made for them if they are not fast enough. This also differs from the real world where researchers can stare at their data as long as they like and then actively decide to collect more. Would it be better to allow subjects to click a button to see more data (or decide between options 1 and 2 given the current data)? This would remove speed of processing/decision making from the picture.

Response: In the course of developing the task, we tried various options, and the format suggested by the reviewer was something we considered. This does have the advantage that the subject would not feel rushed. We agree that with the current version of the task, there is a concern that, while the person is still thinking about possibly responding at a small array size, the next array is displayed. Indeed, we suspect this may be one reason why the mean array index is relatively high, even in block 1, and we don't see the anticipated increase in array index with learning (even though accuracy improves). Quite simply, the task setup discourages responding at small array sizes.

In response to these considerations, we developed a 3rd pilot task and have tried this with 20 participants. In this new version, the participant selects an array size, and a cost is included so that they can earn a bit more if they pick smaller arrays (this is both more naturalistic and deters them from always just selecting the largest array). Results are described in full in the report on OSF (<https://osf.io/s39qd/>); the bottom line is that people don't learn with this approach. This is intriguing - the comments that participants made suggest the problem is that they sometimes get reinforced when picking a small sample size (after all, although power is not high, it is still possible to see clear effects in some runs), and they don't then engage so much with seeing the results from all array sizes. The method with the gif does, we suggest, lead to greater engagement with all the evidence from different sample sizes, and, in part because of the fact that responses have to be made within 2 s, is more likely to lead to experience with array indices of 4 or more. So our impression is that effectiveness of training may depend heavily on exposure to increasing arrays, which demonstrate to participants the volatility of means in small arrays.

We feel that, given the results from the 3 pilots, it is preferable to retain the current format, which is the only version where we have seen significant learning.

2.2. Analysis Plan: "Subjects who score 90% or more correct in block 1 will be removed from the main analysis and treated separately." I would strongly recommend against removing these subjects since regression to the mean becomes a problem. Subjects that score high in Block 1 are those that have a good prior understanding of sample size effects but also those with a "positive residual" -- in other words their observed score is higher than their true understanding because they got lucky. Assume that the training has no effect. When

removing subjects with >90% in Block 1, the Block 4 scores will be higher than Block 1 scores. This occurs because high values are removed from Block 1 thus lowering the mean. And, those with negative residuals in Block 1 regress to the mean and improve, while those with positive residuals in Block 1 cannot regress to lower values since they have been removed. Thus the estimated treatment effect will be a mix of the true treatment effect plus regression to the mean -- both pointing in the same direction. Hence, it's better to retain all subjects and have an uncontaminated treatment effect (the sample size can be increased if there is a loss of power).

If weeding out those who have a good initial understanding of sample size effects is essential, it would be better to use the pretest quiz as a filter, since positive residuals in the quiz will be uncorrelated with positive residuals in Block 1 (but hopefully performance on the quiz correlates with performance in Block 1).

Response: We thank the reviewer for explaining this confound, and we are grateful for the suggestion that we base exclusions on the pretest quiz rather than the initial performance on the training task to avoid regression to the mean. That's an excellent solution, which we have adopted.

2.3. Currently, only data from Blocks 1 and 4 are used for analysis, which ignores 50% of the collected data (Blocks 2 and 3). A simple alternative analysis would be to regress the outcome on block for each subject and calculate the slope of the regression line (outcome for each block vs block number). The collection of slopes could then be tested if they are different from zero (or substituted for "array.diff" in other models). This may not be appropriate if the learning rate across the four blocks is not approximately linear. Such an approach may provide more stable results since information from two additional blocks are used. The pilot data can indicate if this method has merit.

Response: Thanks for this suggestion. We computed slopes as suggested from the pilot 2 data, using percentage correct (which showed learning gains). The slope measures were highly correlated with the difference score (last -first block), with $r = .95$. In addition, one group t -tests were conducted comparing means with zero for (a) difference (block 4 minus block 1) and (b) slopes: results were similar, with the difference score giving a slightly higher t -value. We propose therefore to stick with the difference score, simple though it is, but we will present data on learning curves. Based on the data so far, our impression is that learning is more of a step function than gradual linear increase, though the point at which learning occurs may vary between individuals.

2.4. Would the dollar amount won be a useful outcome variable as it integrates both the correctness and array index information?

Response: We agree this is a good idea. We tried it with our pilot data. The earning score is inevitably highly correlated with the percentage correct ($r = .989$), and shows clear learning between first and last blocks.

2.5 Typos:

Page 11, line 28: Should be "lose 4 points"?

Page 11, line 32: Should be "likelihood of one scenario"?

Response: typos have been corrected.

Appendix B

UNIVERSITY OF OXFORD

Dorothy Bishop FMedSci, FBA, FRS
Professor of Developmental Neuropsychology
Wellcome Principal Research Fellow

Tel: +44 (0)1865 271369
Fax: +44 (0)1865 281255

<http://www.psy.ox.ac.uk/oscci/>
E-mail: dorothy.bishop@psy.ox.ac.uk

**Oxford Study of Children's
Communication Impairments**
Department of Experimental Psychology
Anna Watts Building
Oxford
OX2 6GG

12th November 2021

Dear Editor

I have pleasure in submitting the stage 2 version of our Registered Report:
Can we shift belief in the 'Law of Small Numbers?' to Royal Society Open Science.

Please note the following points:

1. The paper up to the Results section is largely identical to the Stage 1 submission, except for changes in tense of verbs. There are a few other minor changes for clarification - e.g. being explicit about what is pre-registered, labelling the training task as the 'beeswarm task', etc, but we hope these will be acceptable. These were made solely with the goal of helping the reader navigate through what is quite a complex paper.

I do also wonder whether some re-ordering of the preregistered material would be helpful to readers, or possibly dropping some of the sections on pilot data, as these are available on OSF. However, we are happy to take the advice of you and reviewers on this.

2. As discussed previously with the editor, we ran an initial set of 50 participants on the study in a manner that was not in agreement with the pre-registration: this was an oversight, which meant that the quiz items were presented with the two item types blocked rather than randomised. This we refer to as Sample 1.

We therefore ran another 50 participants (Sample 2) with this error corrected. Rather than doing as the editor suggested, and reporting Sample 1 separately as 'exploratory', we did an initial comparison of the quiz data on the two samples, and found compelling evidence that the difference in presentation made no difference to results, with a Bayes Factor showing strong evidence for the null hypothesis of no difference between samples.

We present results separately for the two samples at various points in the manuscript to confirm that they are equivalent, but for the main analysis of the learning task (which was administered in identical fashion to Samples 1 and 2) we combine the two samples to make optimum use of their data. Throughout, we also show that if we rely only on Sample 2, the results are unchanged, apart from the increased uncertainty associated with a smaller sample.

We look forward to hearing from you in due course

Yours sincerely,

Dorothy Bishop, FBA, FMedSci, FRS
Professor of Developmental Neuropsychology

Appendix C

Thanks to the reviewers for their helpful comments. Reviewer 1 had specific suggestions for improving the manuscript, which we have dealt with as follows.

Initial comment

I didn't find Table 8 particularly useful—comparing the “aware” and “unaware” participants on baseline characteristics was not terribly informative, and I would recommend dropping this.

Response: We had wondered about yet another Table in a long paper, but we feel if we omit this, people will wonder whether there were any characteristics of participants that predicted whether or not they learned in the beeswarm game. We have therefore retained this Table, but provided an introductory sentence that provides motivation for the comparison.

2

1. I found the abstract too vague. For example, “there were significant gains on the training.” What does this mean? What improved and by how much? Similarly, “there was no generalisation to understanding of quiz items.” What does this mean specifically? That participants did not have an increase in their quiz score? Readers need some of these details up front in the abstract.

Response: We have revised the Abstract to clarify these points.

2. Tables and figures should stand alone and not require the reader to consult the text. However, almost all the tables and figures contained insufficient information. For example:

-Table 4: Statistics self-rating is on what scale? Understanding T and understanding power are measured on what scale?

Response: Note now added to Table with this information

-Table 5: What is “item type”? Is this P-item versus S-item or S-item versus P-item? What is pre-post? Is this pre versus post or post versus pre? Needs to be clear without having to read the text.

Response: Clearer row labels added

-Table 6: Is very hard to understand without consulting the text. What is “bias”? What is “other1”? What is “other2”? Were these errors from Sample 1 or Sample 2 or both?

Response: Note added to table to clarify.

-Figure 8: How many are in the aware and unaware subgroups? What are the sample sizes? Also, what are the “mean array index”, “d prime”, “absolute log likelihood at point of response”, and “absolute observed effect size at point of response”? This figure again requires the reader to consult the text to understand these outcomes. Some explanation should be given within the figure itself.

Response: The figure caption has been extended to incorporate the explanation of variables from the text.

3. Some text is also hard to follow:

-When describing the pre-training and post-training means on the S-items quiz, I found the descriptions confusing. The authors switch between percentages (e.g., 83%) and decimals (e.g. 0.28), which is confusing. Also, why not present the raw numbers correct rather than the percentages? As a reader, I would find it much clearer and more transparent to learn that the average score was 1.68 out of 6 rather than 28%.

Response: results are now reports in terms of N items correct.

-I also found this description confusing: “The proportion of participants scoring less than 50% on the pre-training P-items was 0.34 for Sample 1 and 0.36 for Sample 2.” Do you mean that 34% scored less than

50%? This is confusing. Why not say, e.g, 34% got less than 3 out of 6 correct on the P-items? Again, focusing on the raw number correct seems much clearer than focusing on the percentage correct.

Response: reworded as suggested

-“We divided the self-reported responses into those that did and did not indicate awareness that larger sample sizes (or longer waiting) gives more reliable estimates.” How many were in each group? This should be made explicit on first mention.

Response: We have now added the numbers as requested.

4. I'd recommend adding a graph (histogram) that shows the distribution of correct answers on the S-items (0-6) before and after the training; or the distribution of the change in number correct (e.g., -1, +1, 0, etc.). This would be a very helpful visualization for the reader to see that quiz scores did not improve.

Response: We have not made this change as the paper already has a large number of tables and figures. The requested information is clear in Table 8, which now has means on pre- and post-test expressed as raw numbers, subdivided according to Awareness subgroup.

Reviewer: 2

Reviewer 2 just spotted one typo, which we have fixed.